# Multi-temporal high-resolution data products of ecosystem structure derived from country-wide airborne laser scanning surveys of the Netherlands

Yifang Shi[*], Jinhu Wang & W. Daniel Kissling

University of Amsterdam, Institute for Biodiversity and Ecosystem Dynamics (IBED), P.O. Box 94240, 1090 GE Amsterdam, The Netherlands

*Correspondence to*: Yifang Shi (y.shi@uva.nl)

**Abstract**

Recent years have seen a rapid surge in the use of Light Detection and Ranging (LiDAR) technology for

characterizing the structure of ecosystems. Even though repeated airborne laser scanning (ALS) surveys

are increasingly available across several European countries, only few studies have so far derived data

products of ecosystem structure at a national scale, possibly due to a lack of free and open-source tools

and the computational challenges involved in handling the large volumes of data. Nevertheless, high-

resolution data products of ecosystem structure generated from multi-temporal country-wide ALS

datasets are urgently needed if we are to integrate such information into biodiversity and ecosystem

science. By employing a recently developed, open source, high-throughput workflow (named

"Laserfarm"), we processed around 70 TB of raw point clouds collected from four national ALS surveys

of the Netherlands (AHN1–AHN4, 1996–2022). This resulted in ~ 59 GB raster layers in GeoTIFF format

as ready-to-use multi-temporal data products of ecosystem structure at a national extent. For each AHN

dataset, we generated 25 LiDAR-derived vegetation metrics at 10 m spatial resolution, representing

vegetation height, vegetation cover, and vegetation structural variability, together with auxiliary data (~

12 GB) such as raster layers of point density, pulse density, flightline timestamp information, terrain and

surface elevation, and masks of water areas, roads, buildings, powerlines and NA values. The data enable

an in-depth understanding of ecosystem structure at fine resolution across the Netherlands and provide

opportunities for exploring ecosystem structural dynamics over time. To illustrate the utility of these data

products, we present ecological use cases that monitor forest structural change and analyse vegetation

structure differences across various Natura 2000 habitat types, including dunes, marshes, grasslands,

shrublands, and woodlands. The provided data products and the employed workflow can facilitate a wide

use and uptake of ecosystem structure information in biodiversity and carbon modelling, conservation

science, and ecosystem management. The full data products are publicly available on Zenodo

(https://doi.org/10.5281/zenodo.13940846) (Shi et al., 2025).

**1 Introduction**

Monitoring ecosystem structure is essential for sustainable forest management (Lindenmayer et al., 2000),

species distribution research (Jetz et al., 2019; Kissling et al., 2018), dynamic ecosystem modelling

(Kucharik et al., 2000), biodiversity monitoring (Noss, 1990), and the conservation and restoration of

terrestrial ecosystems (Ruiz-Jaén and Aide, 2005). As one of the Essential Biodiversity Variables (EBVs)

classes (Pereira et al., 2013), ecosystem structure provides detailed insights into both the vertical and

horizontal profiles of ecosystems, facilitating a deeper understanding of the relationship between

vegetation structure and animal ecology (Davies and Asner, 2014), forest attributes modelling (Coops et

al., 2021) as well as carbon and biomass dynamics (Zhao et al., 2018; Dalponte et al., 2019). However,

until a decade ago, the collection of vegetation structure data was difficult and labour intensive, especially

over large spatial extents. Although previous studies have explored the use of passive remote sensing technologies, such as high-resolution satellite imagery and aerial photographs, alongside field measurements to obtain structural information (e.g. Wolter et al., 2009; Lamonaca et al., 2008), these applications have largely been confined to plot or local scales with limited scalability and uncertain transferability between different regions.

Over the past few decades, the advent of airborne laser scanning has enabled precise and spatially contiguous measurements of ecosystem structural properties such as high-resolution topographic variation and accurate estimation of vegetation height, cover, and canopy structure (Lefsky et al., 2002). The LiDAR technology used in ALS surveys generates discrete returns (point clouds) and/or full-waveform signals by emitting laser pulses from the sensor towards the target objects (e.g. ground, trees, and buildings, etc), recording the distance between the sensor and the objects ("X", "Y", "Z" coordinates), the amount of energy returned to the sensor ("Intensity"), the sequence of returns generated from one pulse ("Return number" and "Number of returns"), the time at which the objects were observed ("GPS time"), and so on. Advances in sensor systems and techniques also allow many countries to carry out ALS campaigns over national or regional extents, producing fine-scale ecosystem measurements across broad spatial extents (Kissling et al., 2022; Assmann et al., 2022). ALS surveys often generate massive amounts of data (e.g. point clouds with a multi-terabyte data volume) which contain ecosystem structural information that is essential for ecological and biodiversity research (Kissling et al., 2022; Koma et al., 2021b; Bakx et al., 2019). Although tools and software for processing large amounts of LiDAR data are increasingly available (Roussel et al., 2020; Isenburg, 2017; Meijer et al., 2020; Kissling et al., 2022; Fischer et al., 2024), significant challenges remain, including the need for specialist expertise, extensive data storage, and substantial computational power (Assmann et al., 2022). Ultimately, ecologists, foresters, biodiversity researchers and land managers require raster layers with vegetation structural information that can be readily integrated into analytical workflows using software that they are familiar with (e.g. GIS, R, Python). Such raster layers, e.g. LiDAR-derived vegetation metrics, are often generated by statistically aggregating the 3D point cloud information within spatial units such as voxels or 2D raster cells (Meijer et al., 2020; Kissling et al., 2022; Fischer et al., 2024). These LiDAR-derived vegetation metrics typically capture three key dimensions of ecosystem structure: vegetation height (e.g. maximum vegetation height, vegetation height at a certain percentile), vegetation cover (e.g. the density of vegetation at a given height layer), and vegetation structural variability (e.g. the vertical or horizontal distribution and variability of vegetation within a spatial unit) (Kissling et al., 2023; Bakx et al., 2019). Providing high-resolution (e.g. 10 m) ready-to-use LiDAR metrics and making them accessible for the public is thus critical for monitoring Essential Biodiversity Variables (EBVs) (Valbuena et al., 2020), modelling species distributions (De Vries et al., 2021; Koma et al., 2021b; Zellweger et al., 2013), and estimating species diversity (Moeslund et al., 2019; Zellweger et al., 2017; Aguirre-Gutiérrez et al., 2017) at a regional or national scale.

Ecosystem structure is a three-dimensional phenomenon with horizontal and vertical components that change over time (Zenner and Hibbs, 2000). The increasing frequency of ALS data acquisition offers a unique opportunity to monitor ecological changes and ecosystem dynamics at fine spatial and temporal scales. Several countries have been conducting repeated (sub-)national ALS surveys to obtain fine-scale information on topography and forest ecosystems (Nilsson et al., 2017). For example, the Dutch national ALS programme (AHN, *Actueel Hoogtebestand Nederland*, https://www.ahn.nl/) has been collecting country-wide LiDAR data since 1996, providing four complete ALS datasets (AHN1–AHN4) with an ongoing fifth survey (AHN5), conducted at intervals of 3 to 5 years. In Spain, under the PNOA-LiDAR project, two national ALS campaigns have taken place during 2008–2015 (LiDAR $1^{st}$ coverage) and during 2015–2021 (LiDAR $2^{nd}$ coverage), while the third acquisition (LiDAR $3^{rd}$ coverage) has started in 2023 and is planned to finish in 2025 (https://centrodedescargas.cnig.es/CentroDescargas/modelos-digitales-elevaciones, last access: 17 May 2025). While the primary goal of many ALS campaigns is to produce terrain and surface elevation models, such as Digital Terrain Models (DTMs) and Digital Surface Models (DSMs), the multi-temporal LiDAR datasets also capture detailed 3D characteristics on vegetation structure over time, providing valuable information for evaluating changes in biomass (Cao et al., 2016; Feng et al., 2024), forest structure (Mccarley et al., 2017; Riofrío et al., 2022; Vepakomma et al., 2011), and forest carbon stocks (Dalponte et al., 2019; Zhao et al., 2018). Furthermore, these datasets are increasingly being integrated with other remote sensing data, such as satellite imageries from Landsat, Sentinel-2, and synthetic aperture radar (SAR), to assess forest changes caused by disturbances like wildfires (Li et al., 2023; Feng et al., 2024) and to model aboveground biomass (Musthafa and Singh, 2022). However, despite the growing availability of multi-temporal ALS datasets, there is a noticeable lack of publicly available data products, i.e. LiDAR-derived vegetation metrics, from national ALS surveys.

Several challenges emerge when generating accurate and standardized data products from multi-temporal ALS data (Valbuena et al., 2020). Over the past three decades, advances in LiDAR sensors and associated technologies have led to improvements in point density, classification accuracy, and additional attributes provided in each point (Riofrío et al., 2022). However, these advancements also introduce complexities in data harmonization. In addition to the challenges associated with processing large datasets and high computational costs (Meijer et al., 2020), discrepancies in sensor technology and flight configurations across different ALS surveys can hinder the generation of consistent data products (Lin et al., 2022). For instance, the first Dutch national ALS campaign (AHN1, 1996–2003) had an average point density ranging from 1 point per 16 square meters to 1 point per square meter, with no detailed point classification available. By contrast, in the fourth campaign (AHN4, 2020–2022), the point density has improved to 20–30 points per square meter, with detailed classification code provided for each point following the ASPRS standard (Asprs, 2019). These technological variations inevitably result in data products with varying quality and accuracy, introducing uncertainties in their usability (Tompalski et al.,

2021; Hopkinson et al., 2008). To understand ecosystem dynamics accurately, changes detected from multi-temporal ALS datasets should reflect actual ecological changes in the target of interest rather than differences in data acquisition or quality (Riofrío et al., 2022). Identifying the limitations and providing usage notes of derived data products are important for users to interpret the data products correctly and apply them optimally in their analyses.

Here, we present a new set of multi-temporal data products of ecosystem structure derived from four national ALS surveys of the Netherlands (AHN1–AHN4). The data products, with a spatial resolution of 10 m, include four sets of 25 LiDAR-derived vegetation metrics representing ecosystem height, vegetation cover, and structural variability, aimed at supporting a wide range of ecological applications. In this paper, we (1) describe the ALS data collection from AHN1–AHN4 and the employed "Laserfarm" workflow to generate the data products, (2) present the detailed characteristics of the generated multi-temporal data products (i.e. LiDAR-derived vegetation metrics as GeoTIFF raster layers) and their known limitations and corresponding usage notes, (3) provide auxiliary data such as raster layers of point density, pulse density, flightline timestamp information, DTMs, DSMs, and mask layers of water areas, roads, buildings, powerlines and NA values to facilitate multi-temporal comparisons, (4) demonstrate two use cases for using the generated data products in ecological applications, and (5) discuss the potential use and recommendations for utilizing these data products in future research. Note that the AHN1 dataset has a rather poor quality, which limits its use for ecological applications. To facilitate open science, we make the data products, employed workflow, Python script, and related documentation publicly available. We anticipate that this will not only allow the upscaling of ecological and biodiversity research but also benefit a broad range of scientists and decision-makers who are interested in using ecosystem structure information for environmental monitoring and management.

## 2 Raw data and processing workflow

### 2.1 Geography and ecology of the Netherlands

The Netherlands is situated in Northwest Europe (52°22′N, 4°53′E), covering a total land area of 33,893 km$^2$. It has mostly flat coastal lowlands and reclaimed land (polders) with an average elevation of approximately 30 meters above sea level. The primary ecosystems in the Netherlands include agricultural land, dunes and beaches, forests, wetlands, grasslands, and other (semi)natural environments (Hein et al., 2020). The Netherlands has a temperate maritime climate with continental influence, resulting in an average annual precipitation of 854.7 mm and a mean temperature of 10.5 ℃.

### 2.2 Four Dutch national ALS campaigns

The initial purpose of the AHN programme was to monitor and manage water systems in the Netherlands. It is a collaboration between 26 regional water boards, provinces and Rijkswaterstaat (the executive

directorate general for public works and water management of the Dutch government) with the aim of producing accurate digital elevation models of the Netherlands. To minimize the impact of foliage on ground detection during the laser scanning, the AHN data acquisition is performed in the winter period, from December to April. The first generation of AHN (AHN1) was conducted during 1996–2003, with a point density of 1 point per 1–16 square meters, which largely depended on the viability of the technology and the date of acquisition (Swart, 2010). Due to errors in the AHN1 data (e.g. inaccuracies in the inertial navigation system, misalignment of overlapping scanning strips, and the presence of artifacts), the data quality of AHN1 is rather poor, especially for areas covered by vegetation (Brand et al., 2003). It is therefore limited in its use for quantifying vegetation structure with high accuracy and at fine (e.g. 10 m) resolution. To support both water and dike management, the second generation of AHN (AHN2) was started in 2007, with improved specifications such as a higher point density (on average 6–10 pts m$^{-2}$) and a higher planimetric/vertical accuracy (5–15 cm). It also required some raster data (i.e. DTMs and DSMs) to be delivered with grid cell sizes of 0.5 m and 5 m. With the main aim of obtaining terrain surface information, both AHN1 and AHN2 datasets were delivered in two separate parts: point clouds representing the terrain ("gefilterde puntenwolk") and point clouds representing non-ground points, i.e. trees, buildings, bridges and other objects ("uitgefilterde puntenwolk").

Benefitting from the advances in LiDAR sensors and related technologies, the third generation of AHN (AHN3) provided not only a higher density of point clouds, but also more information stored for each point, such as point classification code, intensity values, number of returns, and so on (Table 1). Even though both AHN2 and AHN3 were collected within a 6-year cycle (2007–2012 for AHN2, and 2014–2019 for AHN3), the actual time difference between AHN2 and AHN3 varies between 4–10 years depending on the area of interest (Fig. 1). For the latest completed AHN survey (i.e. AHN4), the sampling was conducted between 2020 and 2022 (3-year cycle), making the country-wide dataset more quickly available for the whole Netherlands. All four AHN datasets were provided in LAZ format (i.e. version 1.2 for AHN1–AHN3, and version 1.4 for AHN4), under the local Dutch coordinate system "RD_new" (EPSG: 28992, NAP: 5709). The datasets from AHN1 to AHN4 show an increase in data volume and improved classification as well as additional attributes stored for each point (Table 1). An ongoing fifth ALS survey (AHN5) has started in 2023 (the first part of the data is available, see https://www.ahn.nl/heel-westelijk-nederland-gereed) and the data acquisition will be completed in 2025.

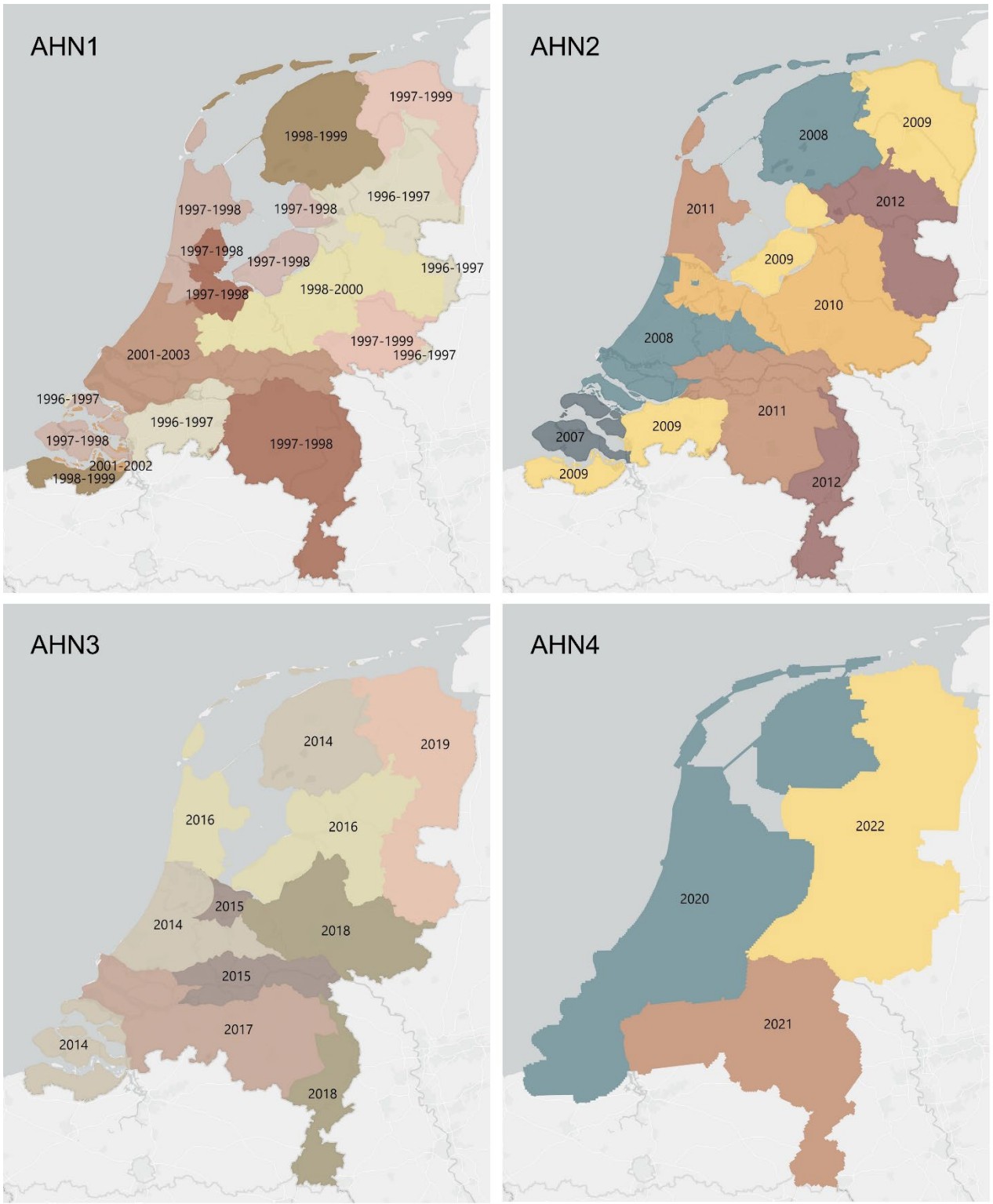

Fig.1 Data acquisition times for AHN1–AHN4. Different colours indicate the different years of data collection for each dataset.

**Table 1.** Summary of raw point cloud characteristics collected by different AHN surveys (AHN1–AHN4). Some flight configurations are not available, for instance, the type of sensor, the flight height, flight speed, and the scan angle, especially for the AHN1 dataset. NAP: Normal Amsterdam Level.

| Data characteristic | AHN1 | AHN2 | AHN3 | AHN4 |
|---|---|---|---|---|
| Acquisition year | 1996–2003 | 2007–2012 | 2014–2019 | 2020–2022 |

| | | | | |
|---|---|---|---|---|
| Acquisition season | Leaf-off | Leaf-off | Leaf-off | Leaf-off |
| Horizontal projection | RD_new | RD_new | RD_new | RD_new |
| Vertical projection | NAP | NAP | NAP | NAP |
| Point density (pts m$^{-2}$) | 0.05–1 | 6–15 | 10–20 | 20–30 |
| Scan angle (°) | - | ± 30 | ± 35 | ± 35 |
| Overlapping rate | - | 20–35% | 20–35% | 20–35% |
| Point cloud format | Laz (1.2) | Laz (1.2) | Laz (1.2) | Laz (1.4) |
| Horizontal accuracy (cm) | - | 8–18 | 8–18 | 8–13 |
| Vertical accuracy (cm) | 5–35 | 5–15 | 5–15 | 5–10 |
| Number of files | 2720 | 60185 | 1367 | 1381 |
| Data volume (compressed) | 33.1 GB | 986.7 GB | 2564.8 GB | 6408.6GB |
| Attributes in each point | X, Y, Z | X, Y, Z | X, Y, Z, intensity, return number, number of returns, classification, scan angle, point ID, GPS time | X, Y, Z, intensity, return number, number of returns, classification, scan angle, point ID, GPS time, amplitude, reflectance, deviation |
| Classification | uitgefilterd (0) gefilterd (0) | uitgefilterd (0) gefilterd (0) | unclassified (1) ground (2) building (6) water (9) reserved (26) | unclassified (1) ground (2) building (6) water (9) powerline (14) reserved (26) |
| Available additional layers | - | DSM, DTM | DSM, DTM | DSM, DTM |

**2.3 Processing workflow**

We employed the high-throughput workflow "Laserfarm" (https://laserfarm.readthedocs.io/en/latest/) to process the multi-temporal AHN datasets. Laserfarm is an open-source workflow designed for processing large amount of LiDAR point cloud data into geospatial data products of ecosystem structure (Kissling et al., 2022). It builds on the feature extraction module of the open-source "Laserchicken" software to compute LiDAR metrics (Meijer et al., 2020). The Laserfarm workflow consists of four main modules: (1) re-tiling, where the original LAZ files (covering 5 km × 6.5 km per tile) are re-tiled into 1 km × 1 km LAZ files for an efficient, scalable and distributed processing; (2) normalization, where a DTM is constructed using the lowest point within a given grid cell (1 m × 1 m), and every point in the cell is then

assigned a normalized height with respect to the derived DTM height, so that the influence of terrain is
removed from subsequent processing. Outliers with z values higher than 10,000 m were removed from
further processing; (3) feature extraction, where user-defined features (e.g. LiDAR metrics such as the
95$^{th}$ percentile of vegetation height and the skewness of vegetation height) are calculated at 10 meter
resolution using points within an infinite square cell (i.e. a 3D square column with a base area of 10 m ×
10 m and an infinite z value) (Meijer et al., 2020); and (4) rasterization, where the extracted feature files
(.PLY files) are merged and exported as single-band GeoTIFF raster files. Note that in all four AHN
datasets, vegetation points are not classified separately based on the ASPRS standard. Instead, they are
assigned a classification value 0 ("uitgefilterd") in AHN1 and AHN2, and a value 1 ("unclassified") in
AHN3 and AHN4. These classification values were used as vegetation class during the feature extraction.
We chose the Laserfarm workflow to process the four country-wide AHN datasets because (1) it enables
the efficient, scalable and distributed processing of multi-terabyte LiDAR point clouds at a national scale,
(2) it is a free and open-source tool implemented in Python and available as Jupyter Notebooks, and (3)
it allows the automated generation of consistent and reproducible geospatial data products of ecosystems
structure from different ALS data.

Due to the different characteristics of each AHN dataset (Table 1), several pre-processing steps
were implemented before executing the main modules of the Laserfarm workflow (Fig. 2). In particular,
for the AHN1 and AHN2 datasets, the step "Reclassification" was carried out before re-tiling, as both
datasets only have "gefilterd" (ground) and "uitgefilterd" (non-ground) files provided and the raw
classification value was set to 0 (never classified) for all points. We therefore reassigned a classification
value "2" to the ground points ("gefilterd") and a classification value "0" to the non-ground points
("uitgefilterd"). These classification values were later used for filtering the points during feature
extraction. Note that there is no publicly available information on the methods/algorithms used in the pre-
classification, and it is therefore difficult to assess the accuracy of the pre-classification of the AHN
datasets. However, a preliminary assessment of the terrain filtering process in the Dutch coastal dunes
did not reveal a strong impact of the ground point pre-classification of AHN datasets on vegetation change
detection (Appendix C). For the AHN4 dataset, the volume of a single original LAZ file varies from 0.3
MB to 16.5 GB, with an average size of 4.6 GB per file (Table 2). Since handling such volumes is
challenging for many computing infrastructures (due to their CPUs and random-access memory, RAM),
we applied a "Splitting" step before the re-tiling (Fig. 2), with a maximum data volume of ~ 500 MB
being used for splitting the original tiles into smaller ones.

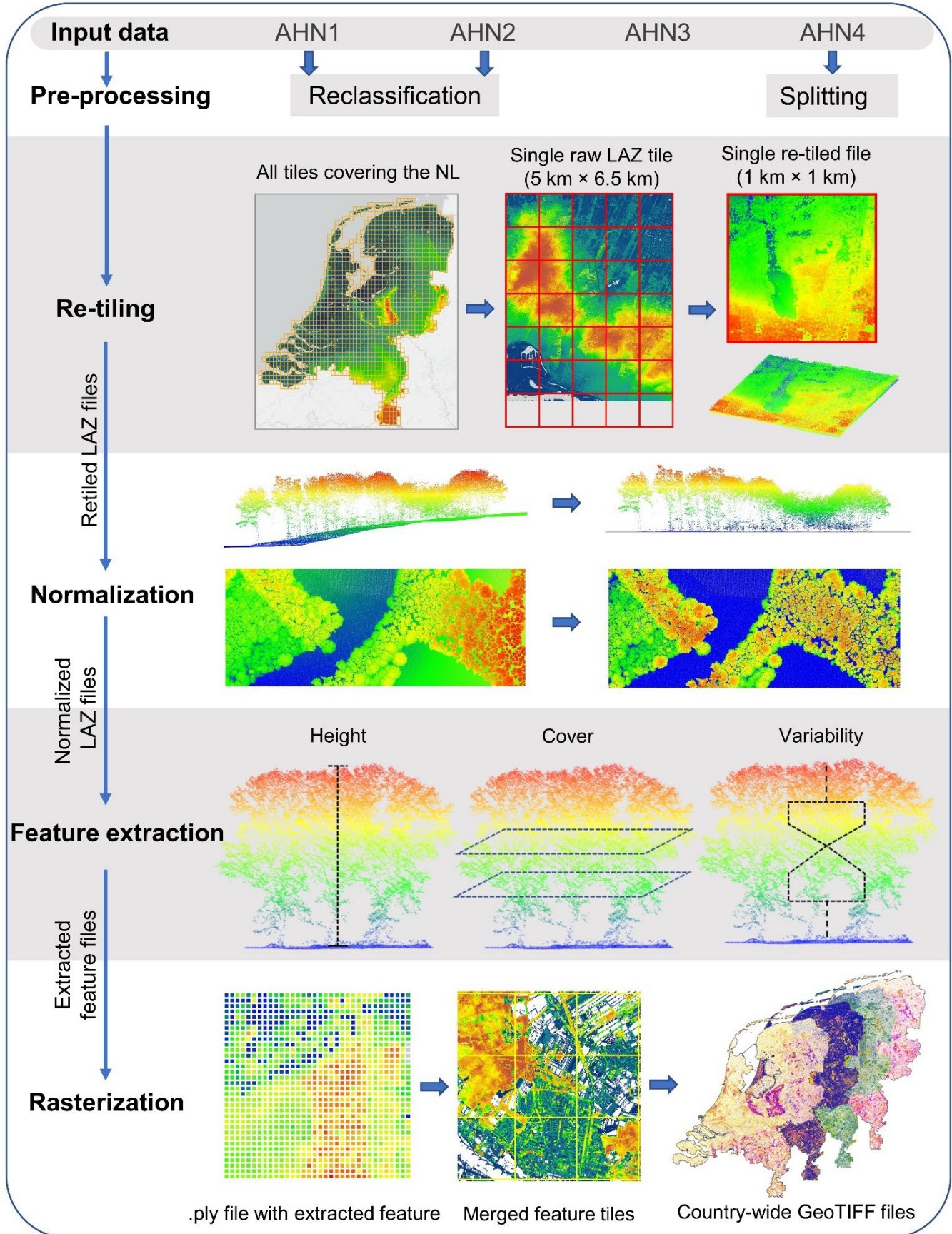

Fig. 2 Overview of the processing workflow employed for four country-wide AHN datasets of the
Netherlands (AHN1–AHN4). The pre-processing step "reclassification" was only conducted for the
AHN1 and AHN2 datasets, where ground points were reassigned a classification value "2". The
"splitting" step was added to split the large LAZ files from AHN4 into smaller ones before re-tiling. Re-
tiling, normalization, feature extraction and rasterization are four main modules of the Laserfarm

workflow, which have been applied for all four AHN datasets to generate country-wide LiDAR-derived vegetation metrics. The input data were raw LAZ files with different point density, and the output data were 25 single-band GeoTIFF raster layers at 10 meter resolution for each AHN dataset.

**2.4 IT infrastructure and computational cost**

All four AHN datasets were processed on the IT infrastructure services provide by SURF, the Dutch national facility for information and communication technology (https://www.surf.nl/). Specifically, we used the dCache platform for data storage (https://www.surf.nl/en/services/dcache) and the HPC Cloud (https://www.surf.nl/en/services/hpc-cloud) or Spider platform (https://www.surf.nl/en/services/high-performance-data-processing) for high-performance data processing. The data processing platforms have fast access to the data storage while enabling scalable and flexible processing of multi-terabytes datasets on distributed resources. We first downloaded the raw AHN1–AHN4 LiDAR point clouds from the PDOK webservices (https://www.pdok.nl/introductie/-/article/actueel-hoogtebestand-nederland-ahn) to the dCache data storage using a customized python script (https://github.com/ShiYifang/AHN/tree/main/AHN_downloading). We then ran the Laserfarm workflow for processing the AHN1–AHN3 datasets on the HPC Cloud, where we set up a cluster of 11 VMs, each VM with 2 cores, 32 GB or 64 GB RAM, and 256 GB local HDD. Due to migration of the computing resources by SURF (from HPC Cloud to Spider), we processed the AHN4 dataset with the Laserfarm workflow on Spider, where a number of flexible and customisable workers with assigned CPU cores were defined based on the computing requirement for each workflow step. We used 2–10 workers, each with 2–4 cores and 16–32 GB RAM for splitting, re-tiling, normalization, and feature extraction, and 2 workers, each with 12 cores and 94 GB RAM for the rasterization step. All input data (i.e. raw LAZ files), intermediate results (e.g. re-tiled LAZ files, normalized LAZ files, featured PLY tiles), and final output (i.e. GeoTIFF raster layers) were automatically stored (and/or retrieved for the next step) on the dCache data storage.

The computing time for each AHN dataset varies based on the input data volume, the required processing steps (Table 2), and the settings of the employed infrastructure. The increase in data volumes from AHN1 to AHN4 resulted in a strong increase of the processing time (Table 2). In total, it required 57.6 days (wall-time) to process the multi-temporal AHN datasets (AHN1–AHN4). The AHN1 (data volume of 33.1 GB) only took a wall-time of 4.8 days to complete whereas the AHN4 (data volume of 6408.6 GB) took a total wall-time of 26.8 days. It is worth noting that the actual computing time of the process might be longer than the wall-time estimates, e.g. due to processing errors, worker failures, and system maintenance.

**Table 2.** Overview of the number of input files, the total volume and the average volume per file for each
processing step, and the total processing wall-time for each AHN dataset. Note that the total wall-time
was estimated based on different infrastructure settings for processing the AHN1–AHN3 (HPC Cloud)
and AHN4 (Spider) datasets.

| Data characteristic | AHN1 | AHN2 | AHN3 | AHN4 |
|---|---|---|---|---|
| *Input for re-tiling* | *(Reclassified)* | *(Reclassified)* | | *(Splitted)* |
| Number of input files | 2720 | 60185 | 1367 | 18797 |
| Total volume | 33.1 GB | 986.7 GB | 2564.8 GB | 6408.6 GB |
| Average volume per file (mean ± SD) | 12.20 ± 10.68 MB | 16.40 ± 14.73 MB | 1.75 ± 0.93 GB | 4.60 ± 2.41 GB |
| *Re-tiling* | | | | |
| Number of re-tiled files | 37715 | 37627 | 37457 | 37990 |
| Total volume | 33.1 GB | 986.7 GB | 2564.8 GB | 6408.6 GB |
| Average volume per file (mean ± SD) | 0.83 ± 1.64 MB | 26.90 ± 35.98 MB | 0.07 ± 0.18 GB | 0.17 ± 0.09 GB |
| *Normalization* | | | | |
| Number of normalized files | 37715 | 37627 | 37457 | 37990 |
| Total volume | 64.0 GB | 3682.4 GB | 6067.5 GB | 9593.3 GB |
| Average volume per file (mean ± SD) | 1.70 ± 2.13 MB | 97.87 ± 59.23 MB | 0.16 ± 0.09 GB | 0.25 ± 0.13 GB |
| *Feature extraction* | | | | |
| Number of featured files | 37715 × 25 | 37627 × 25 | 37457 × 25 | 37990 × 25 |
| Total volume | 257.1 GB | 282.5 GB | 285.9 GB | 212.5 GB |
| Average volume per file (mean ± SD) | 0.29 ± 0.02 MB | 0.30 ± 0.03 MB | 0.33 ± 0.05 MB | 0.23 ± 0.04 MB |
| *Rasterization* | | | | |
| Number of rasterized files | 25 | 25 | 25 | 25 |
| Total volume | 4.8 GB | 19.4 GB | 18.8 GB | 15.6 GB |
| Average volume per file (mean ± SD) | 202.1 ± 101.6 MB | 774.5 ± 303.5 MB | 759.8 ± 226.2 MB | 625.5 ± 160.7 MB |
| *Processing time* | | | | |
| Total processing wall-time (days) | 4.8 | 11.7 | 14.3 | 26.8 |

## 3 Data products description

### 3.1 Overview of data products

The generated data products from each AHN campaign cover the whole Netherlands, ranging from 50.77 °N to 53.36 °N and from 3.57 °E to 7.11 °E. The data products are provided as 10 meter resolution GeoTIFF raster files (25 single-band raster layers for each AHN dataset) in the local Dutch coordinate system "RD_new" (EPSG: 28992, NAP:5709). The total volume of the four sets of 25 LiDAR metrics is approximately 59.2 GB and the total volume of additional masks and auxiliary data is 12.3 GB. The pixel value is stored in 32-bit floating-point precision. The data products are freely accessible via a permanent Zenodo repository (see Sect. 7).

### 3.2 LiDAR-derived vegetation metrics

In total, 25 LiDAR-derived vegetation metrics were generated from each AHN dataset, representing vegetation height, vegetation cover, and vegetation structure variability (Table 3). For vegetation height, we generated 7 LiDAR metrics (i.e. maximum, mean, median, $25^{th}$, $50^{th}$, $75^{th}$, $95^{th}$ percentile of vegetation height) representing the height of vegetation at the canopy surface and for low, middle, and upper vegetation strata (Fig. 3a). For vegetation cover, we derived 11 LiDAR metrics consisting of one metric describing the openness of vegetation (i.e. pulse penetration ratio), one metric describing the density of upper vegetation layer (i.e. canopy cover), and 9 metrics quantifying vegetation density at different height layers (i.e. below 1 m, between 1–2 m, 2–3 m, 3–4 m, 4–5 m, 5–20 m, above 3 m, below 5 m, and above 20 m) (Fig. 3b). The height layers reflect the most relevant height strata to capture the vegetation distribution of major growth forms (e.g. grass, reed, shrubs and trees) (Morsdorf et al., 2010; Miura and Jones, 2010). Special attention was given to represent low vegetation strata (1–5 m) as they are essential for low-stature terrestrial ecosystems such as grasslands, shrublands or agricultural areas when monitoring animal habitats and species distributions (Koma et al., 2021a; Bakx et al., 2019). Note that the pulse penetration ratio is the only LiDAR metric (among the 25 metrics) that used ground points for the calculation. All other 24 metrics are only calculated with vegetation points (i.e. "unclassified" in AHN). For vegetation structural variability, we derived 7 LiDAR metrics representing the vertical variability of vegetation distribution within a cell (Fig. 3c), including the coefficient of variation, Shannon index, kurtosis, skewness, standard deviation, variance, and roughness (sigma) of vegetation height. The detailed description of how those metrics are calculated and their ecological relevance can be found in Table 3.

**Table 3.** Twenty-five LiDAR-derived vegetation metrics capturing ecosystem structure in three key dimensions (vegetation height, vegetation cover and vegetation structural variability), together with their file names in the data products, the formulas for calculation, their descriptions and example of their ecological relevance. Each LiDAR metric is provided as a single-band GeoTIFF raster layer at 10 meter resolution, with the file name "ahn#_10m_xx", where # is the number of AHN campaign ("1–4") and xx is the name of the LiDAR metrics. For instance, "ahn4_10m_ perc_95_normalized_height" represents the 95[th] percentile of vegetation height derived from the AHN4 dataset. For the calculation formulas, $N$ is the total number of normalized vegetation points within a cell, $z_i$ represents all normalized z values in a cell, and $\bar{z}$ is the mean normalized z value in a cell.

| LiDAR metric (abbreviation) | File name (ahn#_10m_xx) | Calculation formula | Description | Ecological relevance |
|---|---|---|---|---|
| *Vegetation height* | | | | |
| Maximum vegetation height (Hmax) | max_normalized _height | $z_{max}$ | Maximum of normalized z within a cell | Height of canopy surface, tree tops |
| Mean of vegetation height (Hmean) | mean_ normalized_heig ht | $z_{mean}$ | Mean of normalized z within a cell | Average height of vegetation, mean tree height |
| Median of vegetation height (Hmedian) | median_ normalized_heig ht | $z_{median}$ | Median of normalized z within a cell | Vegetation height, vertical distribution of vegetation |
| 25th percentiles of vegetation height (Hp25) | perc_25_normali zed_height | $z_{25\ percentile}$ | 25[th] percentile of normalized z within a cell | Density of vegetation in the low stratum |
| 50th percentiles of vegetation height (Hp50) | perc_50_normali zed_height | $z_{50\ percentile}$ | 50[th] percentile of normalized z within a cell. It corresponds to the Hmedian. | Average height and vertical distribution of vegetation |
| 75th percentiles of vegetation height (Hp75) | perc_75_normali zed_height | $z_{75\ percentile}$ | 75[th] percentile of normalized z within a cell | Density of vegetation in the upper stratum |
| 95th percentiles of vegetation height (Hp95) | perc_95_normali zed_height | $z_{95\ percentile}$ | 95[th] percentile of normalized z within a cell | Height of the vegetation canopy surface, avoiding the effect of outliers (compared to Hmax) |
| *Vegetation cover* | | | | |

| Pulse penetration ratio (PPR) | pulse_penetration _ratio | $\dfrac{N_{ground}}{N_{total}}$ | Ratio of number of ground points to total number of points within a cell | Openness of vegetation, canopy fractional cover, laser penetration index |
|---|---|---|---|---|
| Canopy cover (Density_above _mean_z) | density_absolute _mean_ normalized_heig ht | $100 \times \sum [z_i > \bar{z}]/N$ | Number of returns above mean height within a cell | Density of upper vegetation layer |
| Density of vegetation points below 1 m (BR_below_1) | band_ratio_norm alized_ height_1 | $N_{z<1}/N_{total}$ | Ratio of number of vegetation points below 1 m to the total number of vegetation points within a cell | Density of vegetation below 1 m |
| Density of vegetation points between 1–2 m (BR_1_2) | band_ratio_1_nor malized_ height_2 | $N_{1<z<2}/N_{total}$ | Ratio of number of vegetation points between 1–2 m to the total number of vegetation points within a cell | Density of vegetation in 1–2 m layer |
| Density of vegetation points between 2–3 m (BR_2_3) | band_ratio_2_nor malized_ height_3 | $N_{2<z<3}/N_{total}$ | Ratio of number of vegetation points between 2–3 m to the total number of vegetation points within a cell | Density of vegetation in 2–3 m layer |
| Density of vegetation points above 3 m (BR_above_3) | band_ratio_3_nor malized_ height | $N_{z>3}/N_{total}$ | Ratio of number of vegetation points above 3 m to the total number of vegetation points within a cell | Density of vegetation in above 3 m layer |
| Density of vegetation points between 3–4 m (BR_3_4) | band_ratio_3_nor malized_ height_4 | $N_{3<z<4}/N_{total}$ | Ratio of number of vegetation points between 3–4 m to the total number of vegetation points within a cell | Density of vegetation in 3–4 m layer |
| Density of vegetation points between | band_ratio_4_nor malized_ height_5 | $N_{4<z<5}/N_{total}$ | Ratio of number of vegetation points between 4–5 m to the total number of | Density of vegetation in 4–5 m layer |

| | | | | |
|---|---|---|---|---|
| 4–5 m (BR_4_5) | | | vegetation points within a cell | |
| Density of vegetation points below 5 m (BR_below_5) | band_ratio_norm alized _height_5 | $N_{z<5}/N_{total}$ | Ratio of number of vegetation points below 5 m to the total number of vegetation points within a cell | Density of vegetation below 5 m |
| Density of vegetation points between 5–20 m (BR_5_20) | band_ratio_5_nor malized_ height_20 | $N_{5<z<20}/N_{total}$ | Ratio of number of vegetation points between 5–20 m to the total number of vegetation points within a cell | Density of vegetation in 5–20 m layer |
| Density of vegetation points above 20 m (BR_above_20) | band_ratio_20_n ormalized_height | $N_{z>20}/N_{total}$ | Ratio of number of vegetation points above 20 m to the total number of vegetation points within a cell | Density of vegetation in above 20 m layer |

*Vegetation structural variability*

| | | | | |
|---|---|---|---|---|
| Coefficient of variation of vegetation height (Coeff_var) | coeff_var_ normalized_heig ht | $\frac{1}{\bar{z}} \times \sqrt{\sum \frac{(z_i - \bar{z})^2}{N-1}}$ | Coefficient of variation of normalized z within a cell | Vertical variability of vegetation distribution |
| Shannon index (Entropy_z) | entropy_ normalized_heig ht | $-\sum_i p_i \times log_2 p_i$ where $p_i = N_i / \sum_j N_j$, and $N_i$ is the points in bin $i$. | The negative sum of the proportion of points within 0.5 m height layers multiplied with the logarithm of the proportion of points within 0.5 m height layers within a cell | Vertical complexity of vegetation, foliage height diversity |
| Kurtosis of vegetation height (Hkurt) | kurto_ normalized_heig ht | $\frac{1}{\sigma^4} \times \sum (z_i - \bar{z})^4 / N$ where $\sigma$ is the standard deviation of the z value in a cell. | Kurtosis of normalized z within a cell | Vertical distribution of vegetation |

| | | | | |
|---|---|---|---|---|
| Roughness of vegetation (Sigma_z) | sigma_z | $\sqrt{\sum(R_i - \bar{R})^2/(N-1)}$ where $R_i$ are the residual after plane fitting, and $\bar{R}$ the mean of residuals. | Standard deviation of the residuals of a locally fitted plane within a cylinder | Small-scale roughness and variability of vegetation |
| Skewness of vegetation height (Hskew) | skew_ normalized_heig ht | $\frac{1}{\sigma^3} \times \sum(z_i - \bar{z})^3/N$ | Skewness of normalized z within a cell | Vertical distribution of vegetation |
| Standard deviation of vegetation height (Hstd) | std_ normalized_heig ht | $\sqrt{\sum \frac{(z_i - \bar{z})^2}{N-1}}$ | Standard deviation of normalized z within a cell | Vertical variability of vegetation distribution |
| Variance of vegetation height (Hvar) | var_ normalized_heig ht | $\sum \frac{(z_i - \bar{z})^2}{N-1}$ | Variance of normalized z within a cell | Vertical variability of vegetation distribution |

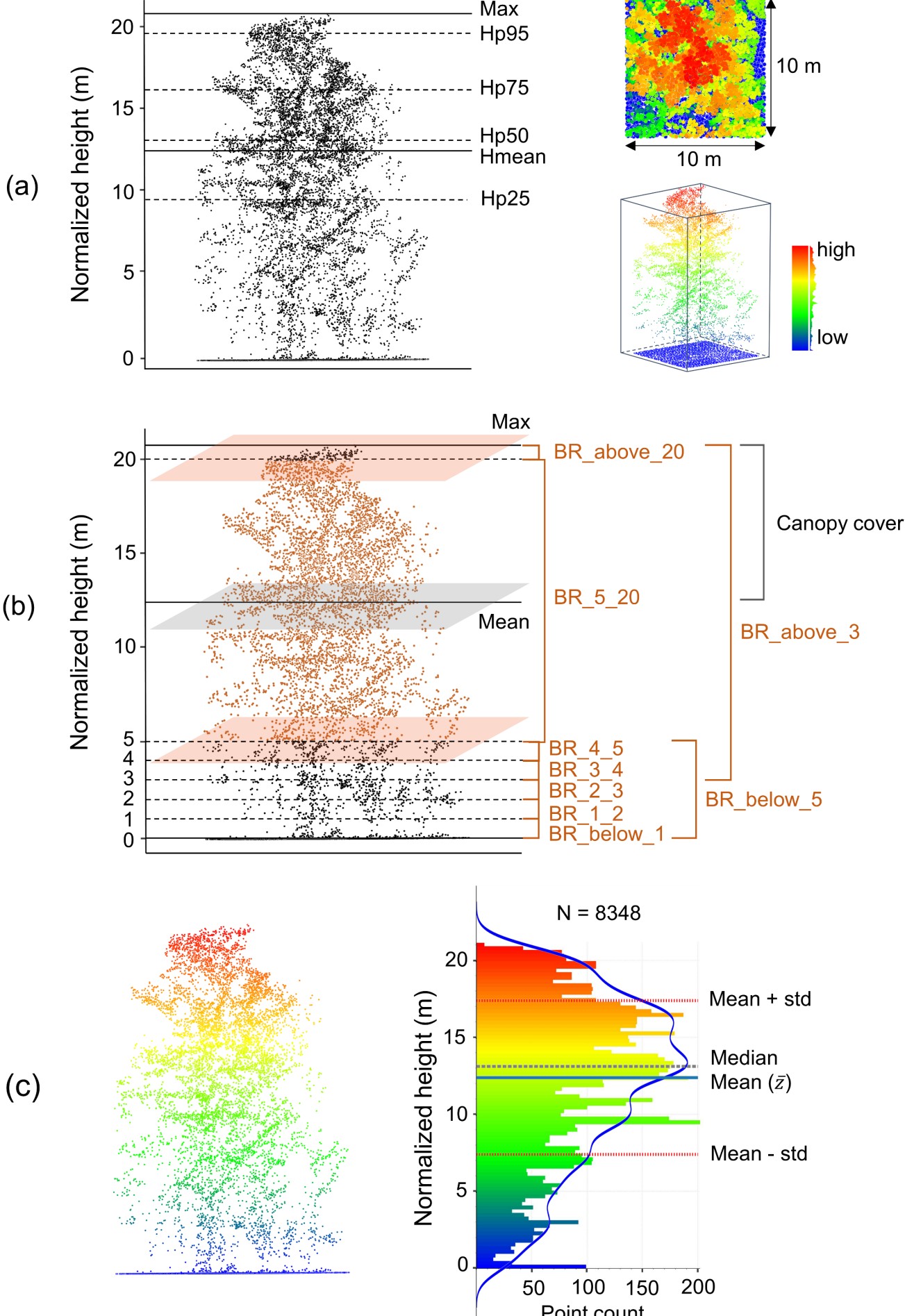

315

Fig 3. Examples of LiDAR metric generation in a 10 m × 10 m grid cell (the number of all points: $N =$ 8348). (a) Metrics of vegetation height (mean, max, and percentiles of normalized height). (b) Vegetation cover metrics representing vegetation density within specific height layers (e.g. "BR_4_5" indicates the vegetation density between 4–5 m, feature name: "band_ratio_4_normalized_height_5"). (c) Metrics of vegetation structural variability (e.g. standard deviation and variance of vegetation height are calculated based on mean height $\bar{z}$; kurtosis and skewness of vegetation height are calculated based on the standard deviation and mean height within a cell) (see detailed calculation formula in Table 3). The blue line in (c) represents a kernel density estimate (KDE) showing the shape of the points distribution. See abbreviation and calculation formula of all metrics in Table 3.

## 3.3 Auxiliary data

Since the point density of AHN datasets changes across space and time, we also provide a raster layer of point density (using all point classes) for each AHN dataset (four in total) (Fig. 4). The AHN1 has a much lower point density (average less than 0.5 pts $m^{-2}$) throughout the whole country than other AHN datasets due to sensor limitations back in 1996. AHN2 and AHN3 have a similar point density (on average 10–20 pts $m^{-2}$), while AHN4 has the highest point density (25–30 pts $m^{-2}$). Especially for the AHN2–AHN4 datasets, distinct patterns (patches, lines, edges) can be observed in different parts of the Netherlands. They are partially due to the influence of the water surface (yellow areas in AHN2, AHN3, and AHN4, Fig. 4), but also related to flight lines and operational configurations (e.g. flying altitude and flight speed) during the campaign.

In addition to point density (i.e. density of all return points), we also provide raster layers of pulse density (i.e. density of first return points) for the AHN3 and AHN4 datasets. Pulse density is less instrument dependent than point density, and reflects more directly the scan quality and condition. Since there is no pulse information available from the AHN1 and AHN2 datasets, we only provide pulse density layers for AHN3 and AHN4. The two pulse density layers are made available in the data repository as auxiliary data together with the derived LiDAR metrics (see Sect. 7).

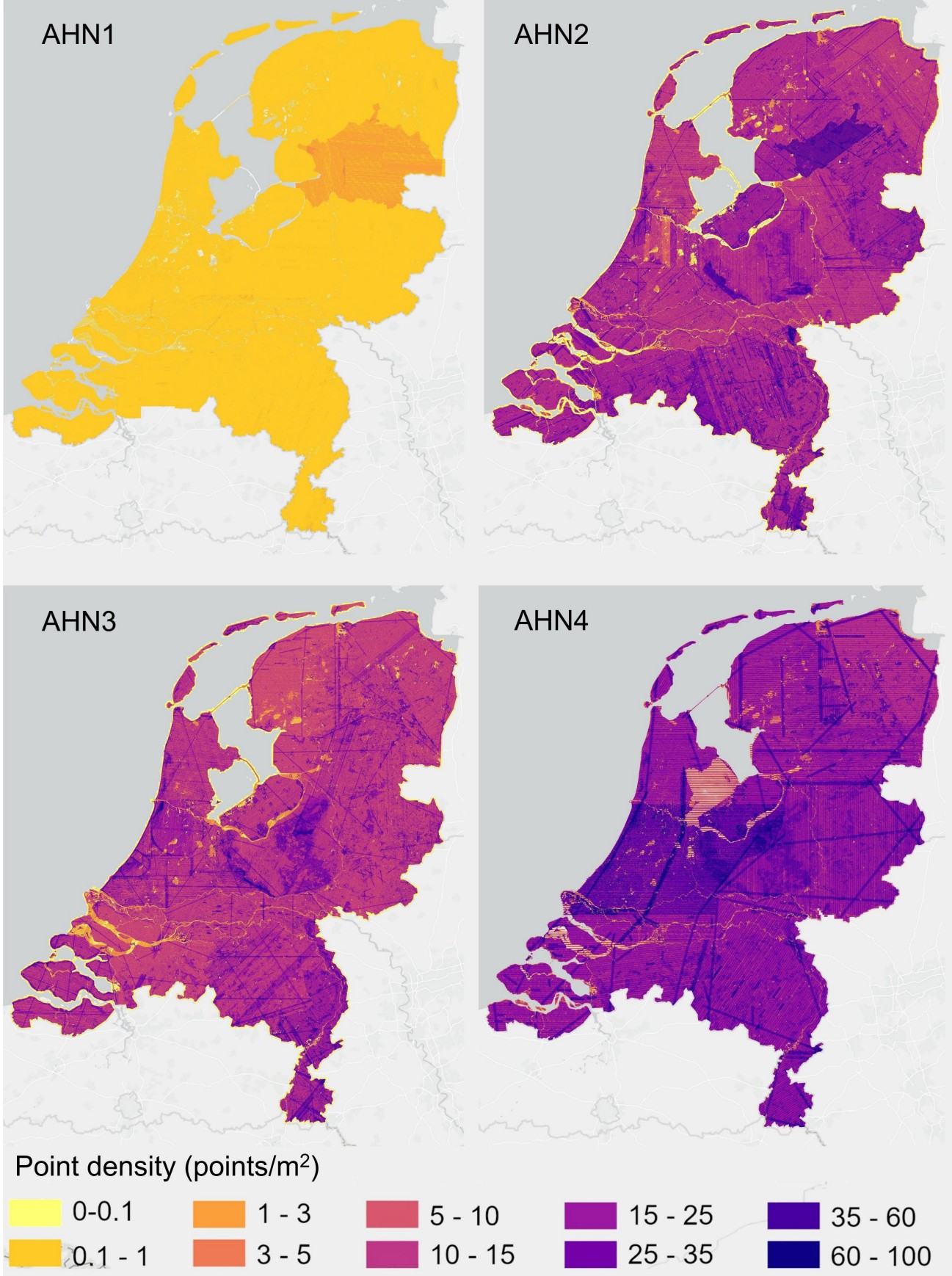

Fig. 4 Point density of AHN1–AHN4 ALS campaigns across the Netherlands. The total number of points was used for calculating the density of points at 10 meter spatial resolution. The four point density layers are made available in the data repository as auxiliary data together with the derived LiDAR metrics (see Sect. 7).

Although AHN campaigns have been conducted during the leaf-off season, the actual date/month that an area has been scanned can vary from December (late winter) to April (early spring), making it difficult to distinguish actual vegetation change (over the years) from leaf phenology. We therefore provide the flightline timestamps as raster layers with a 10 m resolution for comparing the dates of data acquisition across the datasets and generated properties. For AHN3 and AHN4, we first downloaded the flightline vector layers from https://www.ahn.nl/dataroom, and then generated a buffer zone around the flightlines using the function "Buffer" in ArcGIS Pro (version 3.3.0) with the setting of a distance (on both sides of each flightline) of 300 m for AHN3 and 700 m for AHN4. The neighbouring buffer zones were then dissolved if they had the same flight time. The specific distance value of the buffer zone was derived from the distance between two flightlines in each AHN survey. We then rasterized the generated buffer zone polygons into raster layers at 10 m resolution using the "Polygon to Raster" function in ArcGIS Pro. In areas where multiple flightlines are overlapping, we assigned the latest flight date to the raster pixel to be in line with the flight year maps provided by AHN (see Fig. 1). Users should take the surrounding pixel values into account when investigating overlapping areas. The generated timestamp layers for AHN3 and AHN4 are made available in the same data repository as the data products (See Sect. 7 Data availability).

Although AHN provides DTM and DSM layers at 0.5 m and 5 m resolution for AHN2–AHN4, they do not come at the same spatial resolution as the generated LiDAR-derived vegetation metrics. To facilitate users in comparing DTMs and DSMs with the generated LiDAR metrics, we generated DTM and DSM layers at 10 m resolution for each AHN datasets (except AHN1). The generated DTM and DSM layers were derived by resampling DTM and DSM tiles provided by AHN to a 10 m resolution using an unweighted average method. The Jupyter Notebook used for this step is made available in GitHub, see Sect. 6.

**3.4 Limitations and usage notes**

**3.4.1 Classification related errors and masks**

In the pre-classification of the raw AHN point clouds, there is no "vegetation" class provided based on the ASPRS standard (i.e. class 3: low vegetation, class 4: medium vegetation, or class 5: high vegetation). Instead, the vegetation points in the raw AHN1 and AHN2 datasets are included in the non-ground class ("uitgefilterd", classification value of 0), whereas they belong to the class "unclassified" (classification value 1) in the AHN3 and AHN4 datasets (Table 1). This can introduce errors and biases when using the "uitgefilterd" or "unclassified" class for calculating ecosystem structure properties because points belonging to human infrastructures can still be included in these classes. Particularly, buildings and bridges are included (together with other objects other than ground) in the class "uitgefilterd" in the AHN1 and AHN2 datasets, while they are classified separately (buildings in class 6: "buildings", and bridges in class 26: "reserved") in the AHN3 and AHN4 dataset — eliminating the errors caused by buildings and

bridges in the final data products of the AHN3 and AHN4. Powerlines are not separated from the "uitgefilterd" class in the AHN1 and AHN2 datasets, and included in the class "unclassified" in the AHN3 dataset, but in the AHN4 dataset separately classified as class 14: "powerline". Yet, other human objects and infrastructures (e.g. cars, fences, and transmission towers) are not separated in any of the four AHN datasets and thus included in the non-ground class ("uitgefilterd") of the AHN1 and AHN2 datasets and in the class "unclassified" in the AHN3 and AHN4 datasets, introducing some errors and biases in the final data products. There are also points appearing on water surfaces (e.g. reflected by boats and birds) which are included in the class "uitgefilterd" or "unclassified", causing inaccuracies in the final products. In a previous study (Kissling et al., 2023), the accuracy of the 25 LiDAR metrics generated from the AHN3 dataset was assessed, particularly in relation to the error caused by using the class "unclassified" for calculating ecosystem structure properties. The results showed that the overall accuracy of the generated LiDAR metrics was high ($0.90 \pm 0.04$, $n = 25$ LiDAR metrics, tested in 100 randomly selected plots throughout the Netherlands, with 10 m × 10 m size per plot), ranging from 0.87–1. It is worth noting that the impact of those errors on the 25 LiDAR metrics varies, for instance, a stronger bias (i.e. the difference between the generated LiDAR metrics and the ground truth) can be observed in height metrics describing the top canopy layer (i.e. Hmax and Hp95) than in other height metrics or in metrics of vegetation cover in the low strata (i.e. BR_below_1 and BR_below_5) (Kissling et al., 2023).

To minimize the inaccuracies of the data products caused by human infrastructures and water surfaces, we provide mask layers of water areas, roads, and buildings for both the AHN3 and AHN4 data products based on the Dutch cadaster data (TOP10NL) from 2018 (corresponding to AHN3) and 2021 (corresponding to AHN4) (https://www.kadaster.nl/zakelijk/producten/geo-informatie/topnl, last access 18 May 2025). TOP10NL is part of the Basic Topography Registry (BRT) which provides the standard topographic base files for the whole Netherlands. Like the LiDAR metrics, the masks are calculated at 10 m resolution with the RD_new / EPSG 28992 projection coordinate system and provided as raster layers in GeoTIFF format. In the masks, water surfaces, buildings and roads were merged into one class with a pixel value assigned to 1 and the rest with a pixel value of 0 (Fig. 5). Since the historical versions of TOP10NL data are not available for AHN1 (1996–2003) and AHN2 (2007–2012), we can only provide the masks for the AHN3 and AHN4 datasets (see Sect. 7 for data availability). However, despite the potential changes in buildings and roads over time, it is still possible to apply the generated masks to all four AHN data products, for instance, to minimize errors and to have comparable areas of interest. Note that water surfaces were already masked out from the pulse penetration ratio layers by removing 0 values that result from areas with water bodies (i.e. falsely indicating dense vegetation). This was done by masking out water areas (from TOP10NL) from the pulse penetration ratio layers using the "Extract by Mask" function in ArcGIS Pro. Areas with buildings and roads have the value of 1 in the pulse penetration ratio layers which indicates total openness (no vegetation).

Since powerlines are not classified separately for the AHN1–AHN3 datasets and thus included in the vegetation metric calculation, it may cause abnormal values of vegetation structure, especially for vegetation height and vegetation cover above 20 m (Shi and Kissling, 2023). However, in AHN4 the points belonging to powerlines are classified separately (Table 1), which provides a way to minimize errors caused by powerlines in the data products generated from AHN1–AHN3. We therefore extracted all powerline points from the AHN4 raw point cloud and generated a mask (at 10 m resolution) where pixels containing powerlines are assigned a value 1 and the rest as NoData (Fig. 5). Since the transmission towers are not classified separately in all four AHN datasets, the mask only covers the powerlines but not the transmission towers. Users can apply the powerline mask generated from AHN4 to the data products from AHN1–AHN3 and consequently improve the comparability of the LiDAR metrics across time. Note that the powerline infrastructure may also change over time, and the classification of powerlines from the AHN4 may thus not fully represent the powerline distributions in earlier time periods.

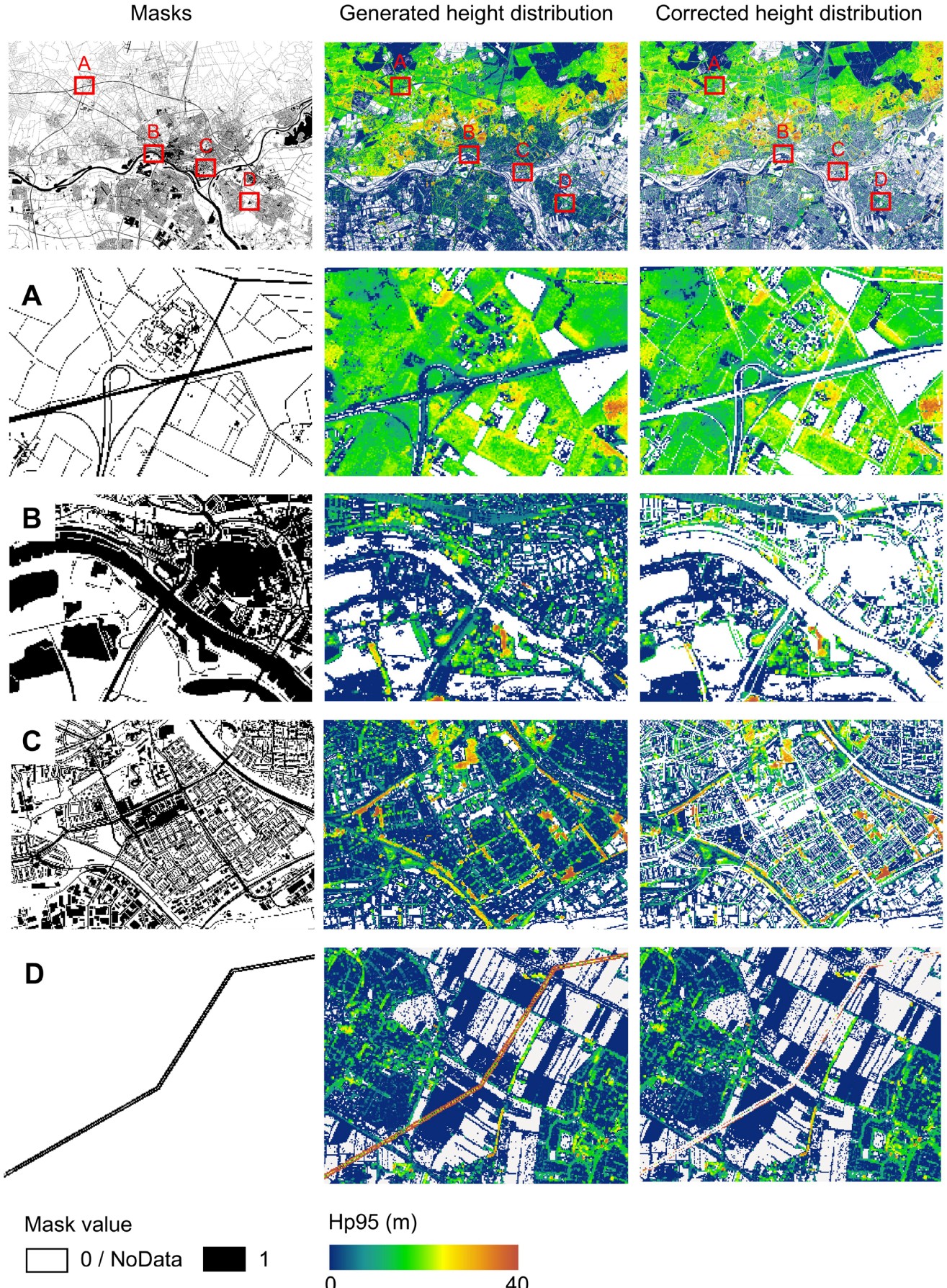

Fig. 5 Examples of masking roads, water surfaces, and buildings as derived from the 2018 Dutch cadaster data (areas A, B, and C) and powerlines generated from the AHN4 (area D). Illustrated is the rasterized mask (first column), the generated vegetation height metric (i.e. Hp95) from AHN3 (second column), and

the corrected vegetation height using the masks (third column). Four subareas show the inaccuracies in the originally generated vegetation height metric and the removal effect of using the mask for roads (area A), water (area B), buildings (area C), and powerlines (area D). A mask value of 1 represents the pixels with roads, water surfaces, buildings, and powerlines, while value 0 or NoData represents the rest. The masks and the LiDAR metrics are at 10 × 10 m resolution. The subareas A–D are located around Arnhem in the east of the Netherlands (5.9102228°E, 51.9825248°N). Hp95 = 95$^{th}$ percentile of vegetation height.

### 3.4.2 Strip issues

Several strip patterns occur in the data products from AHN2 (Fig. 6). This strip issue specifically affects the pulse penetration ratio layer (representing vegetation openness), where both ground points ("ground" class) and vegetation points ("unclassified" class) were used for the metric calculation. A possible reason could be that the scan angle of the laser scanner used for point cloud acquisition was rather wide, and that the scanner thus has received more laser pulses from the areas located at the edges of the flight lines. Those overlapping areas (edges of the flight lines) often have a doubled point density, which also contributes to the strip patterns in the calculation of the LiDAR metrics using ground points (e.g. pulse penetration ratio). This issue most occurs in an area in the centre of the Netherlands (Fig. 6). Some vegetation density metrics (e.g. BR_below_1, BR_below_5) also seem to be influenced by this strip issue.

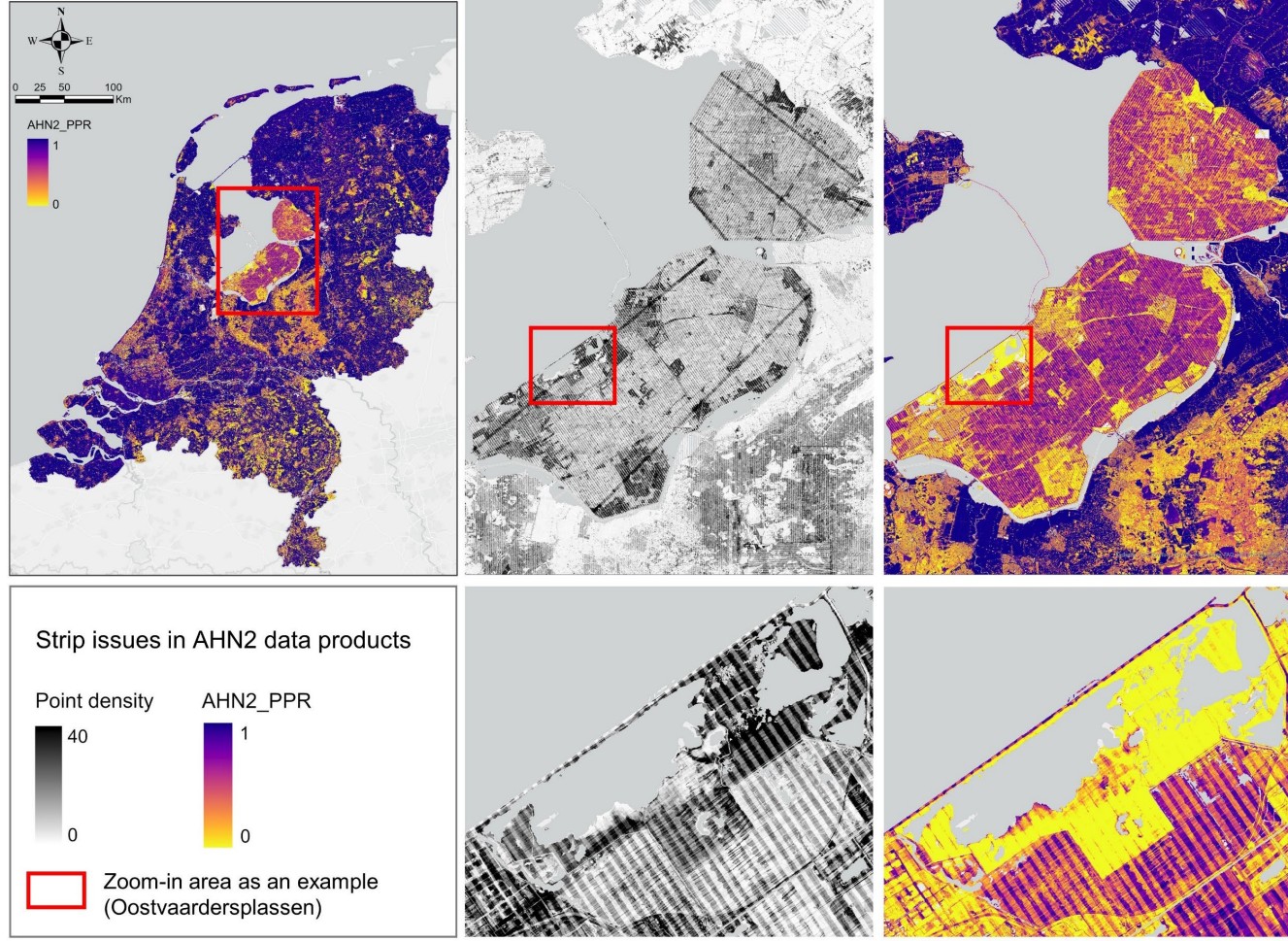

Fig. 6 Strip issues in the AHN2 dataset. The point density (black and white, including all points) and the pulse penetration ratio (colour, representing vegetation openness) show similar strip patterns.

### 3.4.3 Abnormal values

A few pixels with abnormal values still exist in the final products. For instance, several pixels in the Hp95 layer have a value higher than 100 m, which cannot represent the upper canopy of vegetation since the tallest tree in the Netherlands (a Douglas Fir, *Pseudotsuga menziesii*, i.e. a tall and fast-growing conifer native to western North America which was planted between 1860 and 1870 in Apeldoorn, the Netherlands) has been measured to be ~50 meter tall. More generally, most measurements of the tallest trees in the Netherlands range between 20–45 m. Hence, abnormal values of vegetation height (e.g. > 50 m) most likely reflect the occurrence of human infrastructures that are not included in the AHN1 and AHN2 class "uitgefilterd" or not sufficiently captured in the AHN3 and AHN4 classes "building" and "reserved", e.g. aerial and radio masts (up to 350 m tall), tall industrial and meteorological towers and chimneys (50–200 m), cranes (50–130 m), elements of bridges (e.g. pylons and steel cables up to 140 m tall), wind turbines (up to 260 m) and powerlines (up to 80 m). Flying objects, such as birds and planes, can also be captured in the datasets, resulting in abnormal height values in the data products. We recommend filtering out those abnormal values before using the data products for further analysis, e.g. by removing grid cells with $Hp95 > 50$ m, $Hp95 > 40$ m or $Hp95 > 30$ m.

Although the Netherlands has a rather flat terrain, it is worth noting that the normalization method implemented in the Laserfarm workflow may introduce inaccuracies in normalized vegetation height values, especially if steep terrain occurs within a grid cell (Kissling et al., 2022). When applying the same workflow for other countries or regions, abnormal values may occur in areas with drastic topographic changes (e.g. cliffs, mountainous area). Users may consider using a different normalization method, for instance, normalizing non-ground points by subtracting the derived DTM from all points, or by interpolating the elevation of non-ground points using the exact position of ground points beneath (Roussel et al., 2020). Some studies also suggest to use raw point clouds (e.g. the non-normalized DSM) to preserve the geometry of tree tops or plant area index profiles in high slope areas (Khosravipour et al., 2015; Liu et al., 2017).

Since we only used the points from the "unclassified" class of the AHN datasets for calculating vegetation metrics (except for the pulse penetration ratio where all points were used), grid cells with no vegetation points resulted in NA values. Those areas are often bare ground, buildings or water bodies, which should be excluded from vegetation structure assessments. We therefore generated a NA value mask for each AHN dataset (AHN1–AHN4), which can be used for masking areas that have potentially no vegetation (See Sect. 7). Those NA value masks can also be combined and used for vegetation change detection across multi-temporal AHN data products. Note that NA values can also result in areas where very low vegetation is misclassified as ground points, given that the vertical accuracy of the z values in AHN products is typically 5–15 cm (Table 1). Hence, "no-vegetation areas" as derived from the NA value masks can differ from the real land cover.

### 3.4.4 Sensitivity analysis

We conducted a sensitivity analysis to gain a better understanding of the robustness of the LiDAR metrics in relation to the varying pulse densities of the different AHN datasets. We focused on pulse density (i.e. density of the first return points) instead of point density (i.e. density of all return points), as pulse density is less dependent on instrument-specific multiple-return detection capabilities. This makes it more directly related to the scanning parameters (e.g. pulse rate, scanning geometry) and conditions (e.g. flight speed, altitude), reflecting a clearer measure of scan quality. For the four completed AHN surveys, only the AHN3 and AHN4 provide pulse information (e.g. "return number", "number of returns") in the point cloud, whereas the AHN1 and AHN2 does not provide such information. For the latter two, we therefore approximated the pulse information by assuming a pulse density of 1/4 and 1/2 of the AHN3. Since varying pulse density may have different impacts on LiDAR metrics from structurally different habitat types, we performed the sensitivity analysis for five main habitat types (i.e. dunes, marshes, grasslands, shrublands, and woodlands) within Natura 2000 sites in the Netherlands. For each habitat type, 100 sample plots (10 m × 10 m, 500 plots in total) were randomly selected where Hp95 is not NA (assuming vegetation occurring in the plots) (see details of plot selection in Appendix A). For each sample plot, the pulse density of the AHN4 was systematically down-sampled to the same pulse density as AHN3, and then to 1/2 of the pulse density of the AHN3 (assuming comparable with AHN2), and lastly to 1/4 of the pulse density of the AHN3 (assuming comparable with AHN1). For systematic down-sampling, we used the same methodology as described in Appendix B of Kissling et al. (2024a), i.e. the first return points were first sorted according to their GPS acquisition time (from earliest to latest) and then down-sampled to the different densities. For instance, for woodlands, we down-sampled the pulse density from 25 pulses/$m^2$ (AHN4) to 14 pulses/$m^2$, 7 pulses/$m^2$, and 4 pulses/$m^2$, respectively. We then compared the 25 LiDAR metrics generated from the original AHN4 point cloud to those from the down-sampled point clouds for each habitat type. Our analysis revealed that almost all LiDAR-derived vegetation metrics in all habitats are robust to varying pulse densities at 10 m resolution, even when calculated with strongly down-sampled pulse densities of ≤ 4 pulses/$m^2$ (see Figure B1–B5 in Appendix B). The exception were canopy cover ("Density_above_mean_z") and Shannon index ("Entropy_z") which markedly decreased with lower pulse densities in all habitat types, and the coefficient of variation of vegetation height ("Coeff_var") in grasslands and shrublands (see Figure B3–B4 in Appendix B). Some metrics in grasslands also showed larger variability with down-sampled pulse densities.

Given the vertical accuracy of AHN2–AHN4 (i.e. 5–15 cm), classification related errors, and the potential influence of data acquisition time, we suggest that small vegetation changes (e.g. less than 0.5–1 m) should be interpreted with caution. These can be influenced by vertical height uncertainties, low vegetation points being wrongly classified as ground points, or differences in leaf phenology due to varying data acquisition times rather than representing real vegetation changes. When comparing

vegetation changes between the AHN3 and AHN4 metrics, users can make use of the flight time raster layers to take vegetation phenology differences into account. Based on our sensitivity analysis, we also suggest that users should be aware that some LiDAR metrics from open and heterogeneous habitats such as grasslands and shrublands might be less robust to varying point and pulse densities than those from dunes, marshes and woodlands.

# 4 Demonstration of ecological use cases

## 4.1 Monitoring forest structural change across time using multi-temporal ALS data

As a use case, we demonstrate here how the multi-temporal data products generated from the Dutch ALS surveys can capture forest structural change over the past two decades (2000–2023). We included the ongoing ALS campaign (AHN5) since the data were made available for the sample area (central location coordinates: 5.7409230°E, 52.3250517°N) at the time when the analysis was conducted. This provided a longer time series for detecting forest change. The sample area (in a forest area north of the national park De Hoge Veluwe) has experienced a clear forest cut in 2011 (between AHN2 and AHN3 surveys), with further forest loss and some regenerations captured by AHN4, while the latest AHN5 showed a forest regrowth in the middle-low vegetation strata (< 10 m) compared to AHN4 (Fig. 7). Based on the AHN point clouds, the average vegetation height changed from 20.9 m (SD: ± 4.9 m) (AHN1) to 22.6 m (SD: ± 8.0 m) (AHN2), and showed a drastic decrease from 18.0 m (SD: ± 12.1 m) (AHN3) to 3.1 m (SD: ± 4.9 m) (AHN4), and then a slight regrowth to 3.4 m (SD: ± 2.6 m) (AHN5). The histograms derived directly from the AHN1–AHN5 point clouds show the distribution of points shifting from tall vegetation (above 20 m, AHN1–AHN3) to low vegetation (below 10 m, AHN4 and AHN5). Due to the very low point density of the AHN1 data, high-resolution information on vegetation structure in the year 2000 is lacking. However, the histogram from AHN1 implies a similar pattern of canopy height as that from AHN2 (Fig. 7). Google Earth imageries obtained on the closest dates available from each AHN survey also provide a good reference for the forest change events, except for the time of AHN1.

Six selected LiDAR-derived vegetation metrics derived from AHN1–AHN5 at 10 m resolution effectively capture the changes in vegetation structure over time (Fig. 8). The 95th percentile of vegetation height (Hp95) and mean vegetation height (Hmean) highlight reductions in forest canopy height due to cutting in 2011 (between AHN2 and AHN3) and in 2019 (between AHN3 and AHN4). The pulse penetration ratio (PPR) reveals shifts in vegetation openness, with openness peaking in AHN4, while the density of vegetation points at 2–3 m (BR_2_3) indicates regrowth in the understory, particularly in AHN4 and AHN5 (after 2021). The Shannon index (entropy_z) reflects the vertical distribution of vegetation points (i.e. proportion of points within 0.5 m height layers), with AHN2 showing the highest value due to a more even point distribution of the canopy foliage before the canopy was cut. AHN3 shows

the widest Shannon index range, capturing both high canopy trees and new re-growth. The standard deviation (i.e. vertical variability) of vegetation height (Hstd) shows a similar pattern as seen in Hp95.

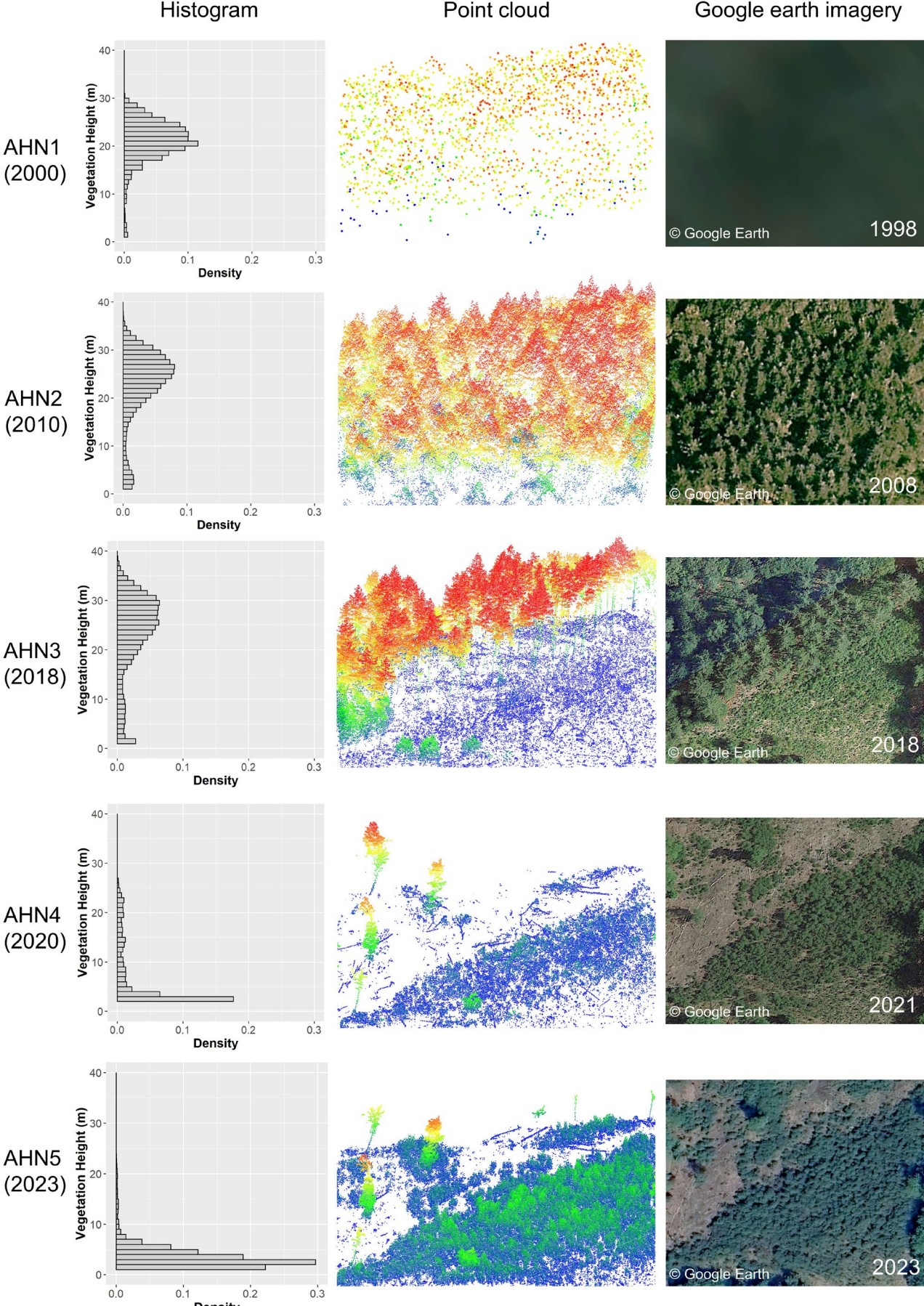

Fig. 7  Forest structural change in a sample plot (100 m × 100 m) between 1998–2023 captured by the
multi-temporal AHN datasets (AHN1–AHN5). The histograms were generated from each AHN point
cloud, showing the distribution of the normalized vegetation height within the plot. The point clouds were
coloured by height (blue indicates lower vegetation height and red indicates higher vegetation height).
AHN1 has a rather poor point density, but shows a histogram of vegetation height that is similar to AHN2.
The forest cut can be observed from the point clouds of AHN3 and AHN4 compared to AHN2, with forest
regrowth occurring in AHN5. Google Earth imageries from the example area show the changes of the
forest. Note that the dates of the Google Earth imageries do not correspond exactly to the dates of the
airborne laser scanning surveys, but to the closest dates available. Map data: © Google Earth.

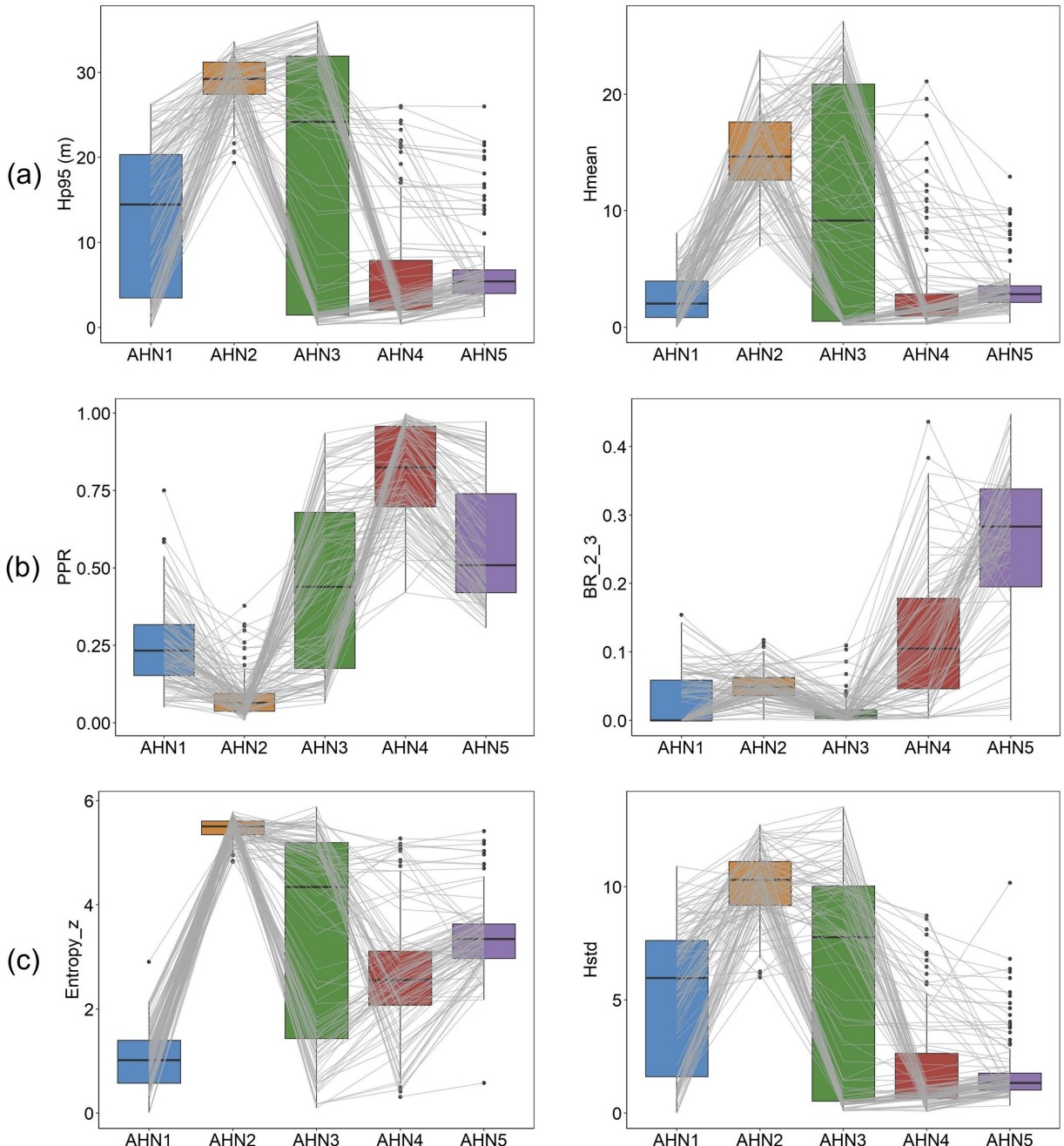

Fig. 8 Boxplots of LiDAR metrics derived from multi-temporal AHN datasets capturing the changes of
the vegetation structure in a 100 m × 100 m sample area (compare Fig. 7). (a) The 95th percentile of
vegetation height (Hp95) and the mean vegetation height (Hmean) representing vegetation height. (b) The
pulse penetration ratio (PPR) and the density of vegetation points between 2–3 m (BR_2_3) representing

vegetation cover. (c) The Shannon index (Entropy_z) and the standard deviation of vegetation height (Hstd) representing vegetation structural variability. Boxes show the median and interquartile range, with whiskers extending to 1.5 times the interquartile range and outliers are plotted as dots. Each grey line represents a single pixel (10 m × 10 m) value changing from AHN1–AHN5, showing the influence of the events on vegetation within each pixel (e.g. forest cut and regrowth).

**4.2 Comparison of vegetation structural difference within Natura 2000 sites**

In a second use case, we analyse how vegetation structure varies spatially across different Natura 2000 habitat types in the Netherlands. Terrestrial habitats were categorized into five main classes: dunes, marshes, grasslands, shrublands, and woodlands, based on the dominant habitat type within each site (see details in Appendix A). For each habitat class, 100 random sample plots (10 m × 10 m, 500 plots in total) were selected where Hp95 is not NA (assuming vegetation occurring in the plots) (Figure A1). We used the data products from AHN4 for the analysis as they are the latest complete products for the whole Netherlands. Four LiDAR metrics were compared: the 95th percentile of vegetation height (Hp95), vegetation point density at 1–2 m (BR_1_2) and 4–5 m (BR_4_5), and the coefficient of variation in vegetation height (Coeff_var). Structural differences among the five habitat types were assessed using the non-parametric Kruskal-Wallis test by ranks (Kruskal and Wallis, 1952), which compares two or more independent groups of equal or different sample sizes without assuming a normal distribution of the residuals. Pairwise comparisons of the statistical significance were conducted among groups (i.e. habitat types) using the Wilcoxon rank-sum test (Wilcoxon et al., 1970).

The strongest structural differences among the five habitat types were observed in canopy height (Hp95) and vegetation density in the lower strata (BR_1_2), followed by vegetation vertical variability (Coeff_var) and vegetation density in the middle strata (BR_4_5) (Fig. 9). Canopy height (i.e. Hp95) of both woodlands and shrublands was highest and showed a statistically significant difference to all other habitat types, whereas grasslands, marshes and dunes did not differ in canopy height (Fig. 9a). The latter three habitat types showed a median canopy height of ~ 2.3 m, whereas it is around 9.9 m and 17.6 m for shrublands and woodlands, respectively. Vegetation density in the low vegetation stratum (between 1–2 m) also did not statistically differ between grasslands, marshes, and dunes (Fig. 9b). However, woodlands and shrublands with their more shaded understory and stronger light competition had proportionally much less vegetation in the lower layer (between 1–2 m) than the three open habitat types (Fig. 9b). In the mid-layer (4–5 m), only the vegetation density of woodlands and marshes showed a statistically significant difference (Fig. 9c). The low mid-layer density in woodlands reflects that understory shrubs are proportionally underrepresented compared to the vegetation density of high canopy trees, whereas shrubs and trees in marshes can be abundant but may generally have a lower canopy height than woodland trees, thus showing high vegetation density at 4–5 m. In terms of structural variability, grasslands and marshes have the highest median values of the coefficient of variation of vegetation height across the 100 plots,

showing significant differences to woodlands, shrublands and dunes (Fig. 9d). This probably reflects a high heterogeneity in vegetation structure in both grasslands and marshes, where a large variability of low vegetation (grasses, herbs) and high vegetation (shrubs, trees) can be present within the 10 m × 10 m plots. It is also the only metric among the four selected metrics where dunes showed statistically significant differences to grasslands and marshes.

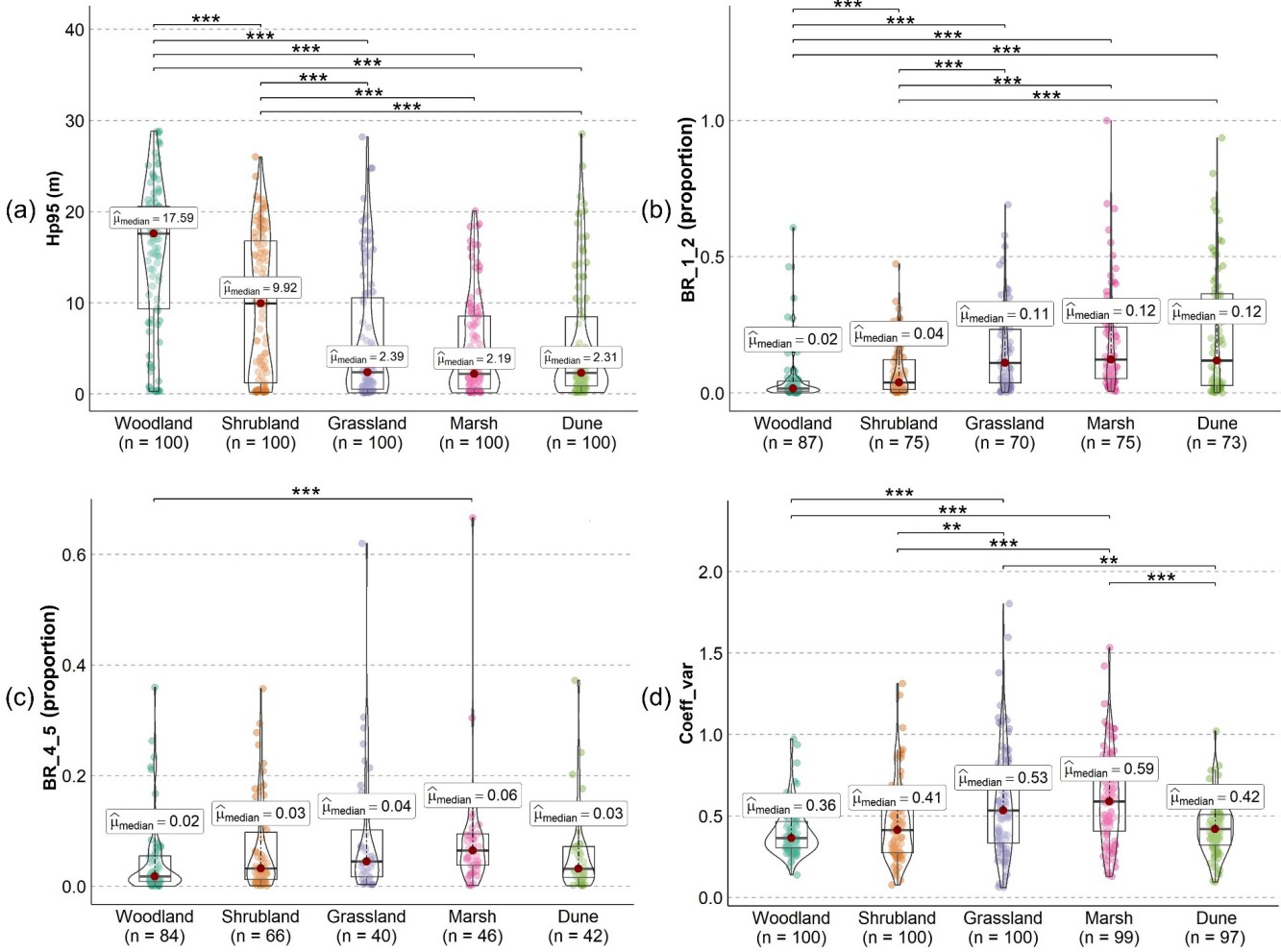

Fig. 9 Comparison of ecosystem structure between five Natura 2000 habitat types using four different LiDAR metrics of vegetation structure. (a) Canopy height (the 95th percentile of vegetation height, Hp95), (b) vegetation density at 1–2 m (BR_1_2), (c) vegetation density at 4–5 m (BR_4_5), and (d) structural variability of vegetation height (coefficient of variation in vegetation height, Coeff_var). The bars above the violin plot indicate whether there is a statistical significance between two compared habitat types. The pairwise comparisons of the statistical significance were conducted using the Wilcoxon rank-sum test after the non-parametric Kruskal-Wallis test by ranks. The significant level is marked as follows: *** ($p < 0.001$), ** ($p < 0.01$), and * ($p < 0.05$). Red dots indicate the median value ($\hat{\mu}_{median}$) of the LiDAR metrics measured for each habitat type. Note that not all sampled plots have vegetation points (from class "unclassified") between 1–2 m and between 4–5 m, therefore the total number of sample plots for the "BR_1_2" and "BR_4_5" analysis was < 100 for each habitat type (after removing NA value). The NA value also occurs for "Coeff_var" when there is only one point (from class "unclassified") in the sampled plot (see metric calculation in Table 3).

# 5 Discussion

We present a set of multi-temporal high-resolution data products of ecosystem structure derived from country-wide ALS surveys of the Netherlands (AHN1–AHN4), capturing vegetation structure dynamics over the past two decades (1998–2022). For each AHN dataset, we provide 25 LiDAR-derived vegetation metrics as GeoTIFF raster layers representing vegetation height, vegetation cover, and vegetation structural variability at 10 m resolution. We further complement these metrics layers with auxiliary data to reduce uncertainties in metric calculations and to facilitate multi-temporal comparisons. In total, we processed ~ 70 TB (uncompressed) raw point clouds from four national ALS surveys into ~ 59 GB GeoTIFF raster layers as final data products, together with auxiliary data (~ 12 GB) including raster layers of point density, pulse density, flightline timestamp information, terrain and surface elevation, and masks of water areas, roads, buildings, powerlines and NA values. These data products hold great value for ecological and geospatial applications, including species distribution modelling, habitat characterization, and forest and biodiversity dynamics monitoring. The availability of these ready-to-use LiDAR metrics enables ecologists and researchers to integrate detailed ecosystem structural information from complex 3D point clouds into their studies without the burden of handling large ALS datasets and computational challenges. Additionally, the dataset serves as a valuable resource for detecting vegetation structural changes and analysing ecosystem dynamics using multi-temporal remote sensing techniques.

Several key aspects should be considered when utilizing the presented data products. First, many commonly used LiDAR-derived metrics, especially those related to vegetation height (e.g. maximum vegetation height, 95$^{th}$ percentile height, mean height), are often highly correlated (Kissling and Shi, 2023; Shi et al., 2018a). To gain a more comprehensive understanding of ecosystem structure, it is advisable to use a complementary set of LiDAR metrics that captures different dimensions of ecosystem structure, or to use dimensionality reduction methods (such as a principal component analysis) to avoid multi-collinearity (Kissling and Shi, 2023). For instance, using the coefficient of variation of vegetation height (Coeff_var) instead of the standard deviation (Hstd) as a metric of structural variability can avoid correlations with mean or canopy vegetation height (Hmean and Hp95) (Kissling and Shi, 2023). Second, vegetation cover in different height layers is a crucial component of forests and other ecosystems, influencing energy fluxes between the ecosystem and the atmosphere (Shugart et al., 2010; Toivonen et al., 2023). Unlike the cover metrics proposed by Moudrý et al. (2022), where herbaceous, shrub and tree layers were used to represent different vegetation strata, our metrics use fixed height intervals (e.g. 1–2 m, 2–3 m, 3–4 m, 4–5 m, 5–20 m, above 20 m) to ensure applicability across diverse ecosystems. Not all ecosystems share the same vegetation growth forms, making these height bin-defined metrics more ecosystem-agnostic. The cover metrics from different height layers can be used as predictors of species distributions (Davies and Asner, 2014), plant diversity (Coverdale and Davies, 2023) and habitat characteristics (Vierling et al., 2008; Bakx et al., 2019). Third, LiDAR metrics related to vegetation

structural variability (e.g. Hstd, Hskew, and Hkurt) are often influenced by various ecological and sensing methodology-related factors, making them potentially challenging to interpret (Assmann et al., 2022). However, metrics representing structural variability are valuable input for models assessing forest functional diversity and structural types (Atkins et al., 2023), especially when combined with optical remote sensing (Kamoske et al., 2022; Zheng et al., 2021). Thus, careful selection of LiDAR metrics for specific applications is highly recommended. Terrain and surface descriptors such as DTMs and DSMs (or canopy height model as derivative) can be additionally considered because they are important for forest and habitat classifications (Shoot et al., 2021), quantifying soil moisture or wetness (Assmann et al., 2022), and analysing species composition (Toivonen et al., 2023; Hill and Thomson, 2005).

While multi-temporal ALS data offer valuable insights into fine-scale vegetation structural changes and ecosystem dynamics, there are also notable challenges, especially when performing change detection and spatial comparisons across point clouds with different characteristics, such as point/pulse density, scanning angle, and varying vertical and horizontal accuracy (White et al., 2016; Kissling et al., 2024a). Instead of performing change detection directly on point clouds (Xu et al., 2015; Kharroubi et al., 2022), many studies use rasterized LiDAR metrics for monitoring changes on vegetation structure. This is computationally less intensive and better suited for areas with complex vegetation structure as it regularizes complex 3D point cloud information onto a 2D grid (Vastaranta et al., 2013; Choi et al., 2023). Several commonly used change detection methods can be applied to the multi-temporal data with rasterized LiDAR metrics. These include image differencing (i.e. subtracting the pixel values of one raster layer, such as Hp95 from AHN3, from the other, such as Hp95 from AHN4), threshold-based change detection (i.e. classifying the pixels as "changed" or "unchanged" based on a set threshold after image differencing), and post-classification comparison (i.e. comparing classified raster layers, such as maps of vegetation types based on derived LiDAR metrics, from different time periods) (Noordermeer et al., 2019; Dalponte et al., 2019). Those methods can be applied to the provided AHN data products, especially after masking water areas, roads, buildings, powerlines, and NA values. Change metrics derived from multi-temporal LiDAR data can also be combined with clustering methods to characterize areas of structural changes, such as modifications of forests by the eastern spruce budworm (Trotto et al., 2024). Together with the development of deep learning on change detection (Bai et al., 2023), more in-depth insights from the presented AHN datasets can be revealed, enabling accurate and comprehensive analysis of ecosystem dynamics. Given the consistent coordinate system used in the four AHN datasets (EPSG: 28992, NAP: 5709; see Table 1), additional georeferencing steps are unnecessary before conducting further analysis with the data products that we provide. The scan angle, overlapping rate, and vertical accuracy of AHN2–AHN4 are rather comparable (Table 1), potentially reducing errors related to systematic differences across time. However, the data products are generated from point clouds with different point and pulse density, which may introduce inconsistencies in capturing vegetation structure. However, our sensitivity analyses showed that most of the vegetation metrics calculated at a 10 m resolution are robust in relation to changes

in pulse density, even when down-sampled to pulse densities of $\leq 4$ pulses/m$^2$. This was largely consistent across different habitat types. Exceptions are canopy cover ("Density_above_mean_z") and the Shannon index ("Entropy_z"), and to a lesser extent the coefficient of variation of vegetation height ("Coeff_var"), especially in grasslands and shrublands. Low vegetation (e.g. in grasslands and dunes) is generally prone to be misclassified as ground points and a low pulse and point density can influence normalization and feature extraction. We therefore recommend that temporal vegetation changes of < 0.5–1 m should be carefully explored, e.g. by using the provided auxiliary data of point density, pulse density, and flightline timestamp information. Still, several studies indicate that the spatial distribution of the point cloud remains similar with variation in point density and that increases in point density do not necessarily increase area-based estimation accuracy (Hudak et al., 2012; Fekety et al., 2015; Cao et al., 2016). We therefore anticipate that the data products from AHN2, AHN3, and AHN4 are reliable for a careful change detection. However, due to the low point density and reduced accuracy, we do not recommend including the data products from AHN1 in multi-temporal analysis.

All software and tools employed in the pipeline for producing the data products are free and open-source, ensuring a standardized yet flexible processing framework for country-wide ALS data and enabling reproducibility for future surveys. While existing ALS processing software such as OPALS (Pfeifer et al., 2014) and LAStools (http://lastools.org/) are not (fully) open-source, and others like FUSION (https://forsys.sefs.uw.edu/fusion/fusionlatest.html), CloudCompare (https://www.danielgm.net/cc/), and lidR (Roussel et al., 2020) lack horizontal scalability and are not specifically designed for processing large ALS datasets on cloud infrastructures with reproducible end-to-end workflows, the employed "Laserfarm" workflow fills a niche by addressing these challenges. Laserfarm is a high-throughput, modular, and reproducible end-to-end workflow designed for efficiently extracting LiDAR metrics of ecosystem structure using distributed computing infrastructures (Kissling et al., 2022). With the workflow materials that we provide, users can implement additional pre-processing steps (e.g. splitting, reclassification) and customize required parameters based on the input ALS data and available computing resources. The demonstrated configurations of IT infrastructure, computational cost, and time efficiency for processing multi-temporal AHN datasets serve as a reference for users to estimate the processing requirements for future national or regional ALS datasets. It is worth noting that the normalization method implemented in the Laserfarm workflow subtracts the elevation of the lowest point within a given neighbourhood to remove the influence of the terrain. This approach was specifically chosen for its effectiveness in handling small ditches and canals that are common in the Dutch landscape, providing a straightforward way to generate positive height values after normalization. However, it may be less suited for capturing continuous normalized height values and fine-scale terrain variability in smaller grid cells (< 1 m) (Kissling et al., 2022). For complex terrains and mountainous areas, both ground classification and terrain model derivation remain challenging and could lead to uncertainties in the generation of vegetation structure properties.

The data products presented here also make a great contribution to multi-source data fusion in remote sensing and ecological research (Ghamisi et al., 2019). Through the two use cases in Sect. 4, we demonstrate the utility of these multi-temporal datasets for monitoring long-term forest dynamics and characterizing habitat types. These applications can be further extended to other studies, such as improving land cover classification accuracy, particularly for objects composed of similar materials (e.g. grasslands, shrubs, and trees). Moreover, the fusion of vegetation structural information from LiDAR, spectral data from optical remote sensing (e.g. high-resolution digital aerial photogrammetry, Landsat and Sentinel-2 imagery), climate data, and field measurements underscores the value of integrating complementary remote sensing data across diverse applications. These include wildlife habitat characterization (Boelman et al., 2016), tree species identification (Shi et al., 2018b), forest structure and carbon stock mapping (Li et al., 2024), as well as assessing disturbances and recovery of ecosystem process (Li et al., 2023). Additionally, combining ecosystem structure data from multiple LiDAR platforms, such as terrestrial, drone-based, airborne, and spaceborne LiDAR, could provide a more comprehensive understanding of ecosystem structure, spanning from understory to canopy level and across local plots to national or continental level.

## 6 Code availability

Jupyter Notebooks for processing AHN datasets: https://github.com/ShiYifang/AHN

Laserfarm workflow repository: https://github.com/eEcoLiDAR/Laserfarm

Laserchicken software repository: https://github.com/eEcoLiDAR/laserchicken

Code for downloading AHN dataset: https://github.com/ShiYifang/AHN/tree/main/AHN_downloading

Code for generating masks for AHN datasets: https://github.com/ShiYifang/AHN/tree/main/AHN_masks

Code for demonstration of ecological use cases: https://github.com/ShiYifang/AHN/tree/main/Use_case

## 7 Data availability

All data products from AHN1–AHN4 (25 GeoTIFF layers for each AHN dataset), six DTM and DSM layers (for AHN2–AHN4), seven masks (two for roads, water surfaces, and buildings from both AHN3 and AHN4, one for powerlines generated from AHN4, and four for NA values for AHN1–AHN4), four point density layers (for AHN1–AHN4), two pulse density layers (for AHN3–AHN4), and two flight timestamp layers (for AHN3–AHN4) are available from a Zenodo repository (https://doi.org/10.5281/zenodo.13940846) (Shi et al., 2024). The data used for the demonstrated use cases are also provided in the same repository. A detailed description of the provided data can be found in the README file in the data repository.

# 8 Conclusions

Ecosystem structure information derived from country-wide ALS data becomes increasingly needed for biodiversity science and ecosystem monitoring. The multi-temporal data products of ecosystem structure and the employed workflow presented here not only provide ready-to-use information for ecosystem monitoring and modelling within the Netherlands, but also enable reproducing desired data products from existing and upcoming large-scale ALS data beyond the Netherlands. We highlight the capability of multi-temporal ALS data products in capturing ecosystem structural dynamics across time and their usability in combination with other data sources. We also carefully evaluated the limitations and usability of generated data products and provided solutions or recommendations for future processing and usage. We envisage that the provided data products and the employed workflow will empower a wider use and uptake of ecosystem structure information in biodiversity and ecosystem science, land management, natural resource conservation, and policy support and decision making.

# Appendix A

The source information about Natura 2000 sites was retrieved from the Europe Environment Agency (Natura 2000 (vector) - version 2021). The shapefile of the Natura 2000 sites and the attributes of each site that we used for the analysis were downloaded via https://sdi.eea.europa.eu/datashare/s/JWt9KJCFMrPQDc7/download. The information on the habitat class (from the table named "Natura2000_end2021_HABITATCLASS.csv") was used to group them into five habitat types (i.e. dunes, marshes, shrublands, grasslands, and woodlands). The table contains the following information: description of the habitat class, habitat code, site code, and percentage of habitat composition within the site.

We first selected all the Natura 2000 sites within the Netherlands (i.e. SITECODE starting with NL), then summarized the highest percentage of habitat class within each site and grouped them into six main habitat types: water, dunes, marshes, shrubland, grassland, and woodland. For water, we included marine areas, sea inlets (habitat code: N01), tidal rivers, estuaries, mud flats, sand flats, and lagoons (habitat code: N02), and inland water bodies (habitat code: N06). For dunes, we included costal sand dunes, sand beaches, and machair (habitat code: N04). For marsh, we included bogs, marshes, water fringed vegetation, and fens (habitat code: N07) and salt marshes, salt pastures, and salt steppes (habitat code: N03). For shrubland, we included heath, scrub, maquis and garrigue, and phygrana (habitat code: N08). For grassland, we included dry grassland, steppes (habitat code: N09), humid grassland, mesophile grassland (habitat code: N10), and improved grassland (habitat code: N14). For woodland, we included broadleaved deciduous woodland (habitat code: N16), coniferous woodland (habitat code: N17), evergreen woodland (habitat code: N18) and mixed woodland (habitat code: N19). For each Natura 2000 site, the habitat type with the highest composition percentage was chosen as the dominate habitat. In total, there were 197 Natura 2000 sites within the Netherlands, including 36 water sites, 25 dune sites, 23 marsh sites, 17 shrubland sites, 54 grassland sites, and 42 woodland sites. For our study, we excluded water sites for the vegetation structure analysis (remaining 161 sites in total). For each habitat type, we randomly selected 100 sample plots (10 m × 10 m for each plot, i.e. in total 500 plots) where Hp95 is not NA (assuming vegetation occurring in the plots) using the *sampleRandom()* function in R (Figure A1). The shapefile of the 500 sample plots across the Natura 2000 sites was then used to extract the pixel values of the LiDAR metrics for comparison.

The shapefile of the Natura 2000 sites within the Netherlands (with habitat class information in attributes), 100 sample plots for each habitat class, original and grouped habitat class information (.csv files), and the R processing script are provided in the data repository (see Sect.7).

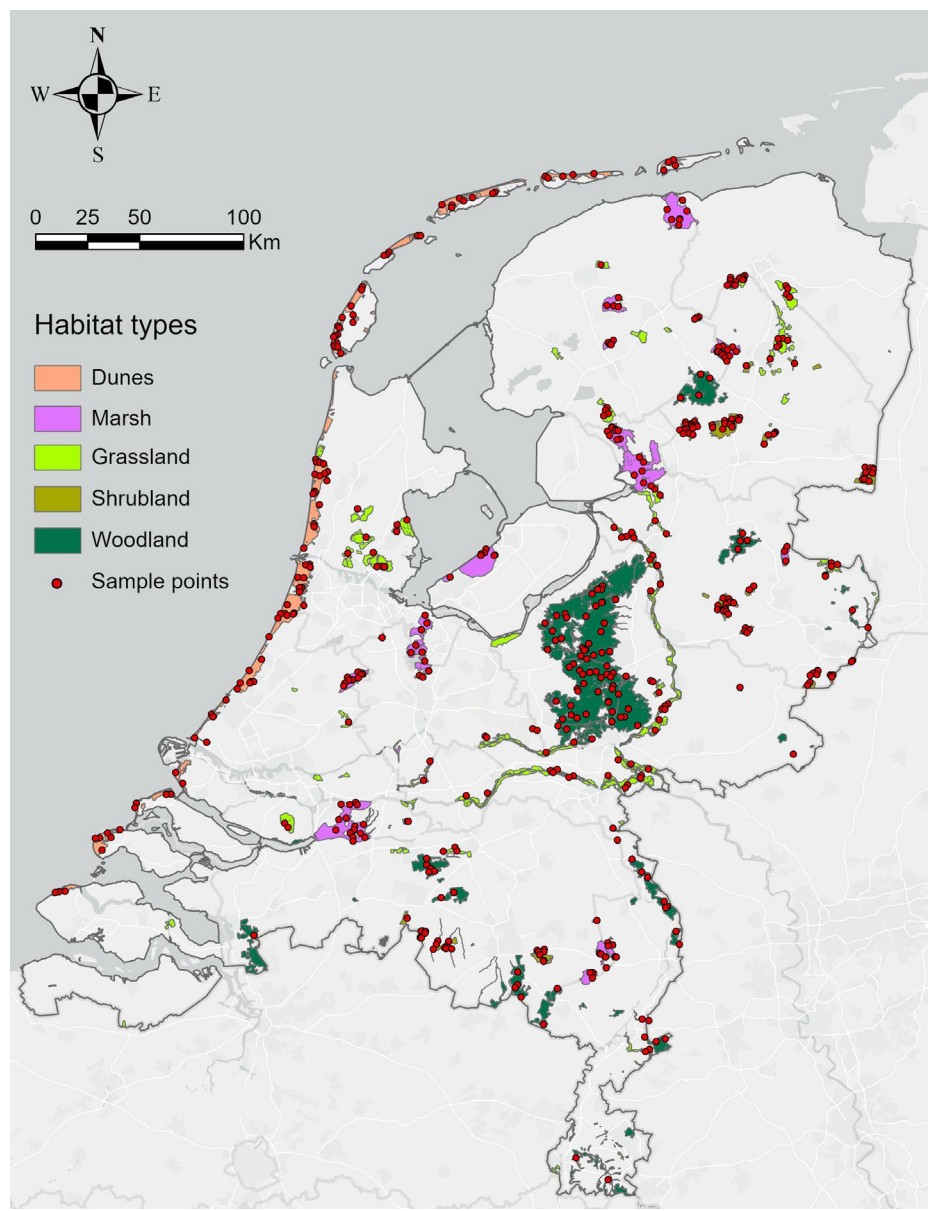

Figure A1. Natura 2000 sites and their habitat types in the Netherlands. The non-water habitat types were
grouped into 5 classes (i.e. dunes, marshes, grasslands, shrublands, and woodlands) to conduct vegetation
structure comparisons. For each class, we randomly sampled 100 plots (10 m × 10 m each) where Hp95
was not NA (assuming that vegetation occurs in the plots) for the analysis ($n$ = 500 in total).

## Appendix B

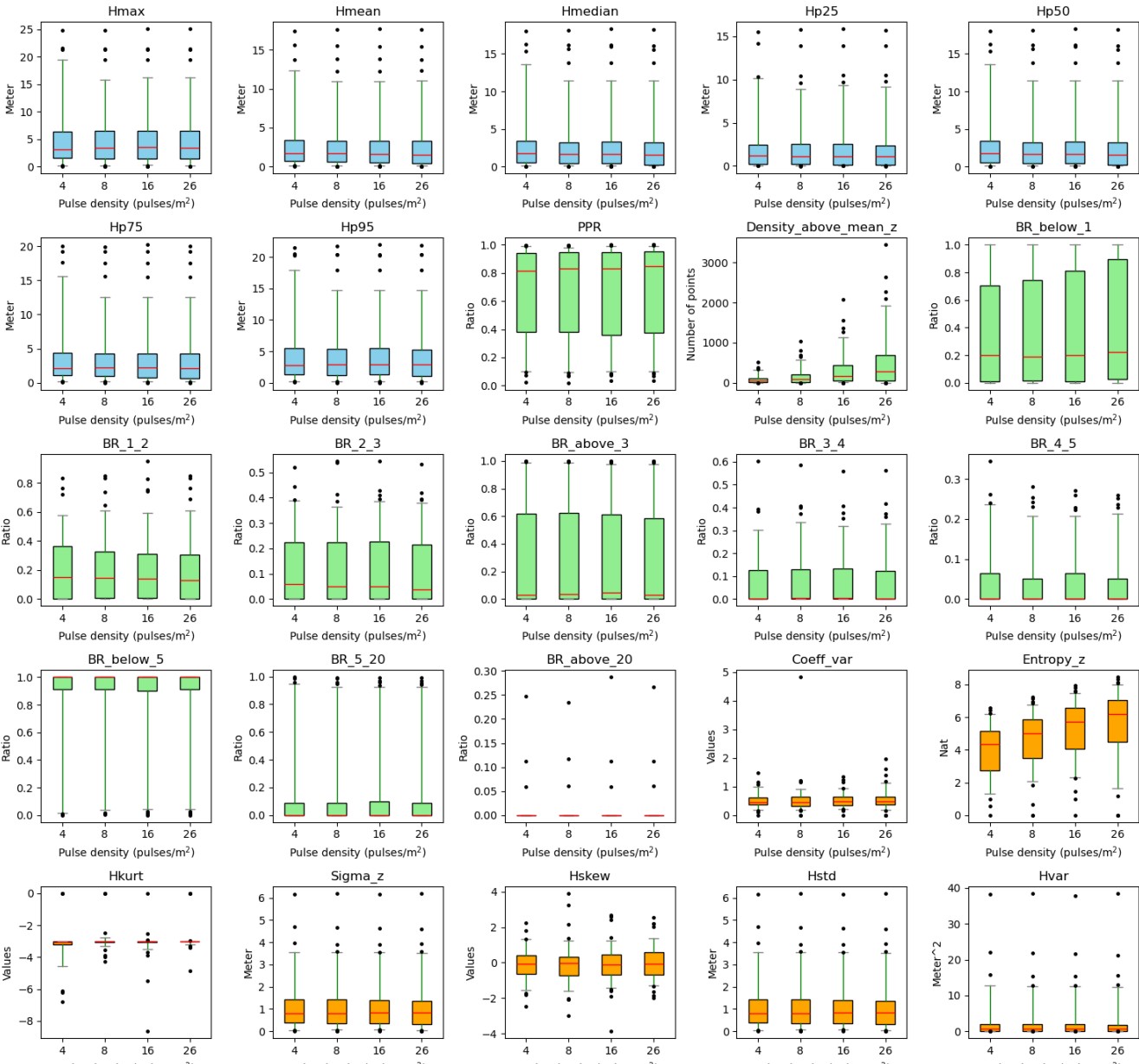

Figure B1. Robustness of vegetation metrics in dune habitats. Twenty-five LiDAR metrics (blue: vegetation height metrics, green: vegetation cover metrics, orange: vegetation structural variability metrics) were calculated with different pulse densities across 100 plots of 10 × 10 m resolution in dune habitats in the Netherlands. Pulse densities were systematically down-sampled based on their GPS time from the original AHN4 dataset to the pulse density of AHN3 and two lower pulse densities (i.e. 1/2 and 1/4 of the pulse density of AHN3 to represent AHN2 and AHN1, respectively). Boxes represent the interquartile range, horizontal red lines the medians, whiskers extend to the 5th and 95th percentiles, and outliers are plotted as dots. See Table 3 for metric explanations.

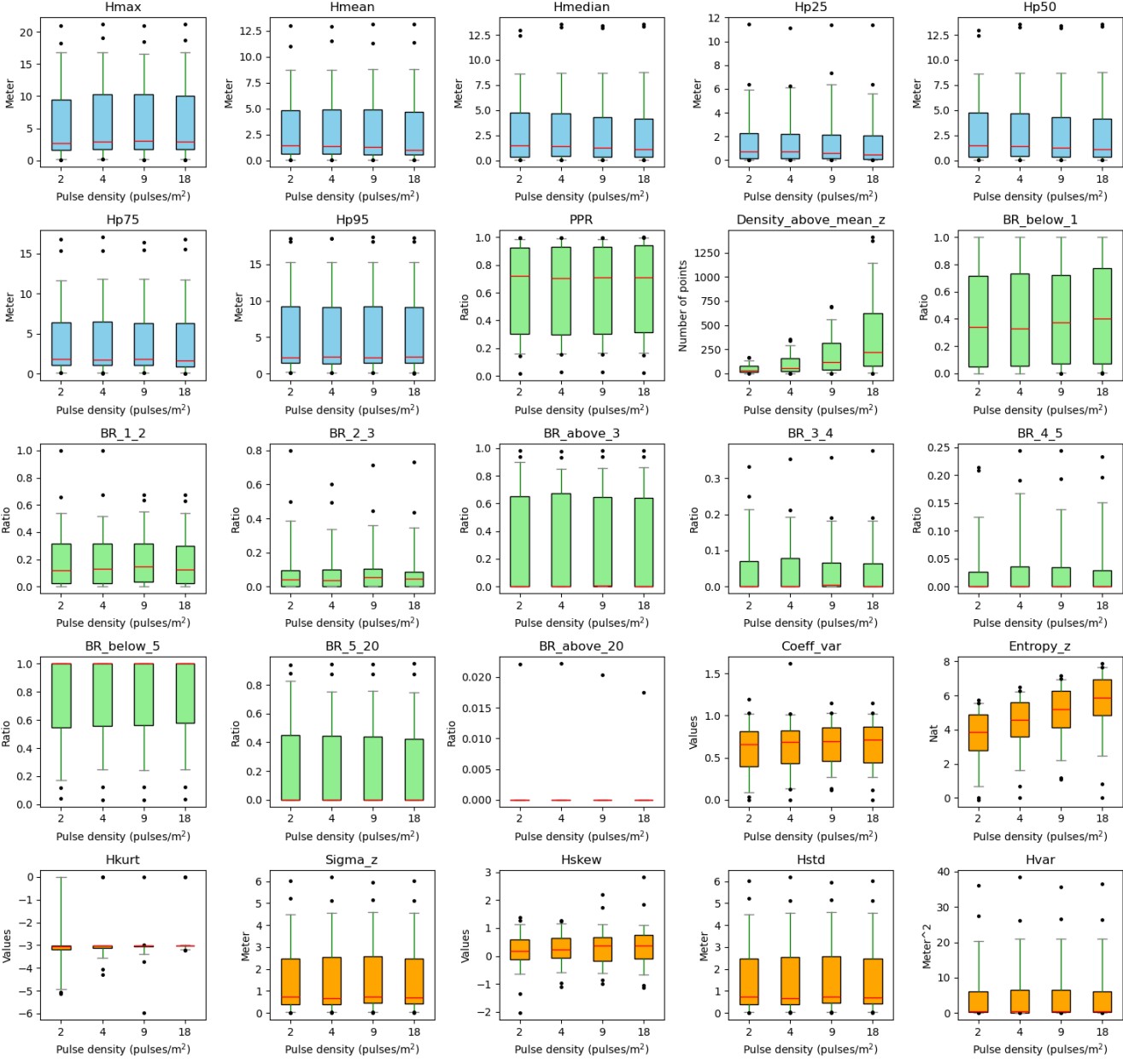

LiDAR-derived vegetation metrics of marsh at different pulse densities

Figure B2. Robustness of vegetation metrics in marsh habitats. Twenty-five LiDAR metrics (blue: vegetation height metrics, green: vegetation cover metrics, orange: vegetation structural variability metrics) were calculated with different pulse densities across 100 plots of 10 × 10 m resolution in marsh habitats in the Netherlands. Pulse densities were systematically down-sampled based on their GPS time from the original AHN4 dataset to the pulse density of AHN3 and two lower pulse densities (i.e. 1/2 and 1/4 of the pulse density of AHN3 to represent AHN2 and AHN1, respectively). Boxes represent the interquartile range, horizontal red lines the medians, whiskers extend to the 5th and 95th percentiles, and outliers are plotted as dots. See Table 3 for metric explanations.

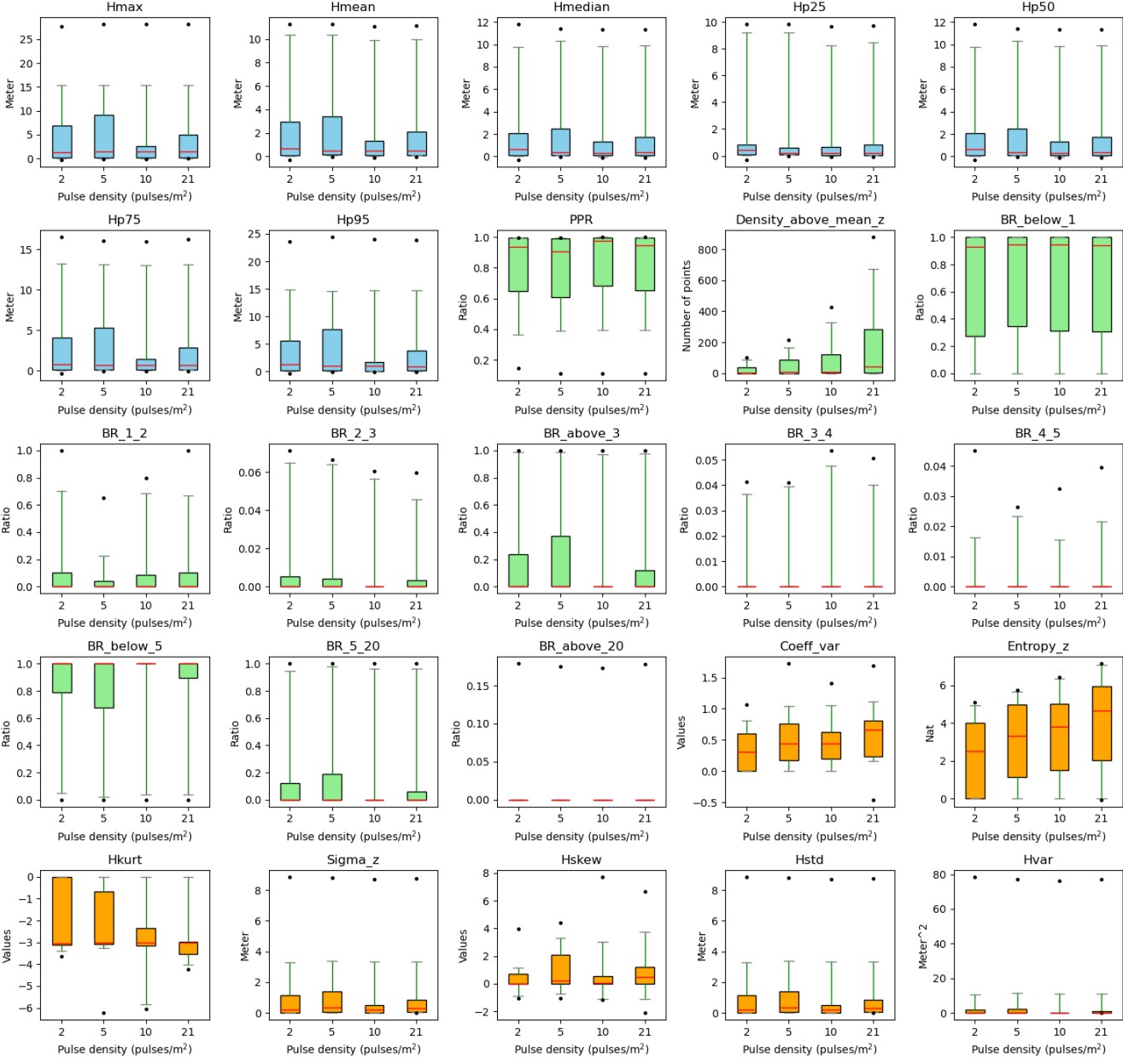

Figure B3. Robustness of vegetation metrics in grassland habitats. Twenty-five LiDAR metrics (blue: vegetation height metrics, green: vegetation cover metrics, orange: vegetation structural variability metrics) were calculated with different pulse densities across 100 plots of 10 × 10 m resolution in grassland habitats in the Netherlands. Pulse densities were systematically down-sampled based on their GPS time from the original AHN4 dataset to the pulse density of AHN3 and two lower pulse densities (i.e. 1/2 and 1/4 of the pulse density of AHN3 to represent AHN2 and AHN1, respectively). Boxes represent the interquartile range, horizontal red lines the medians, whiskers extend to the 5th and 95th percentiles, and outliers are plotted as dots. See Table 3 for metric explanations.

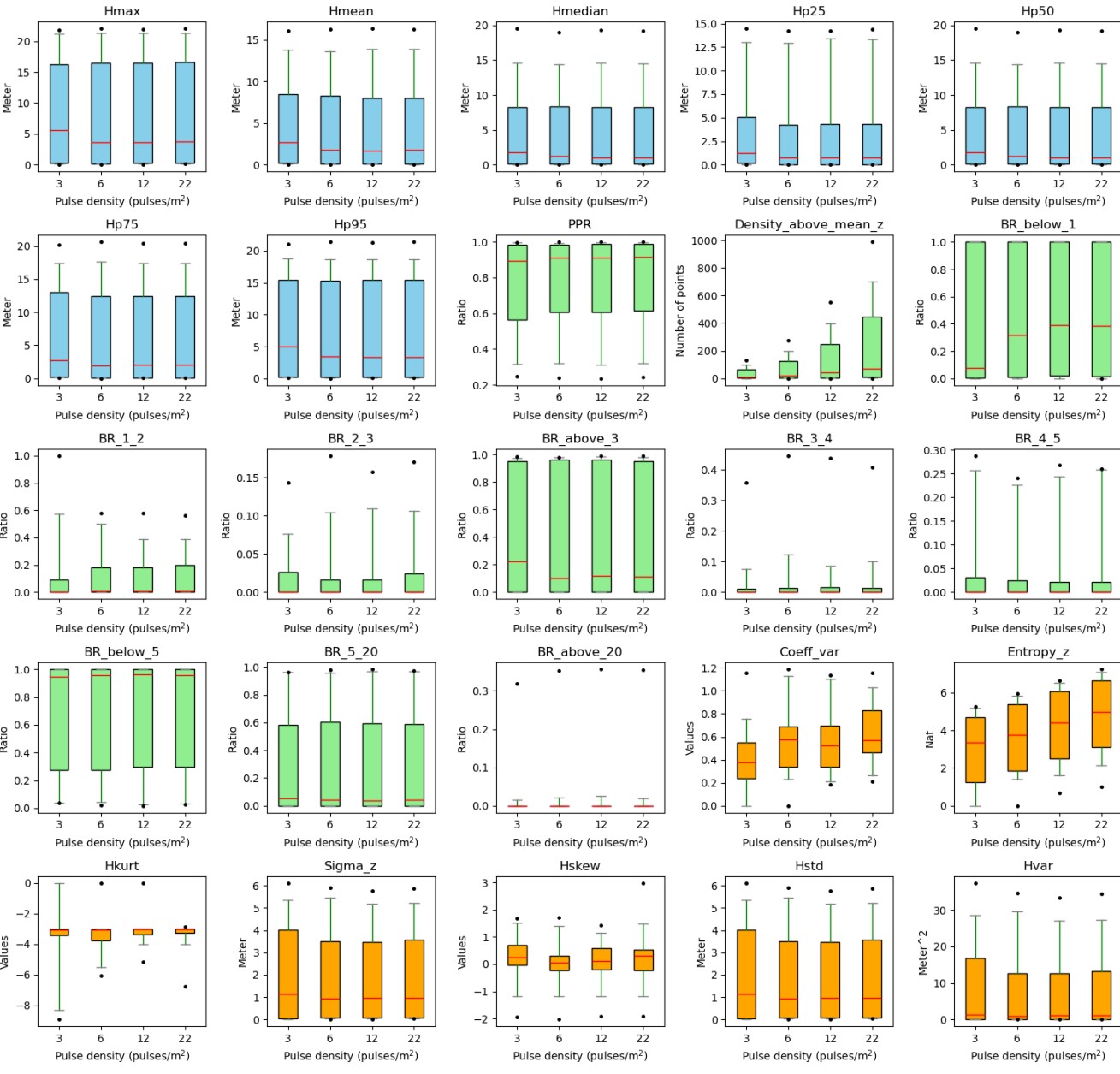

LiDAR-derived vegetation metrics of shrubland at different pulse densities

Figure B4. Robustness of vegetation metrics in shrubland habitats. Twenty-five LiDAR metrics (blue: vegetation height metrics, green: vegetation cover metrics, orange: vegetation structural variability metrics) were calculated with different pulse densities across 100 plots of 10 × 10 m resolution in shrubland habitats in the Netherlands. Pulse densities were systematically down-sampled based on their GPS time from the original AHN4 dataset to the pulse density of AHN3 and two lower pulse densities (i.e. 1/2 and 1/4 of the pulse density of AHN3 to represent AHN2 and AHN1, respectively). Boxes represent the interquartile range, horizontal red lines the medians, whiskers extend to the 5th and 95th percentiles, and outliers are plotted as dots. See Table 3 for metric explanations.

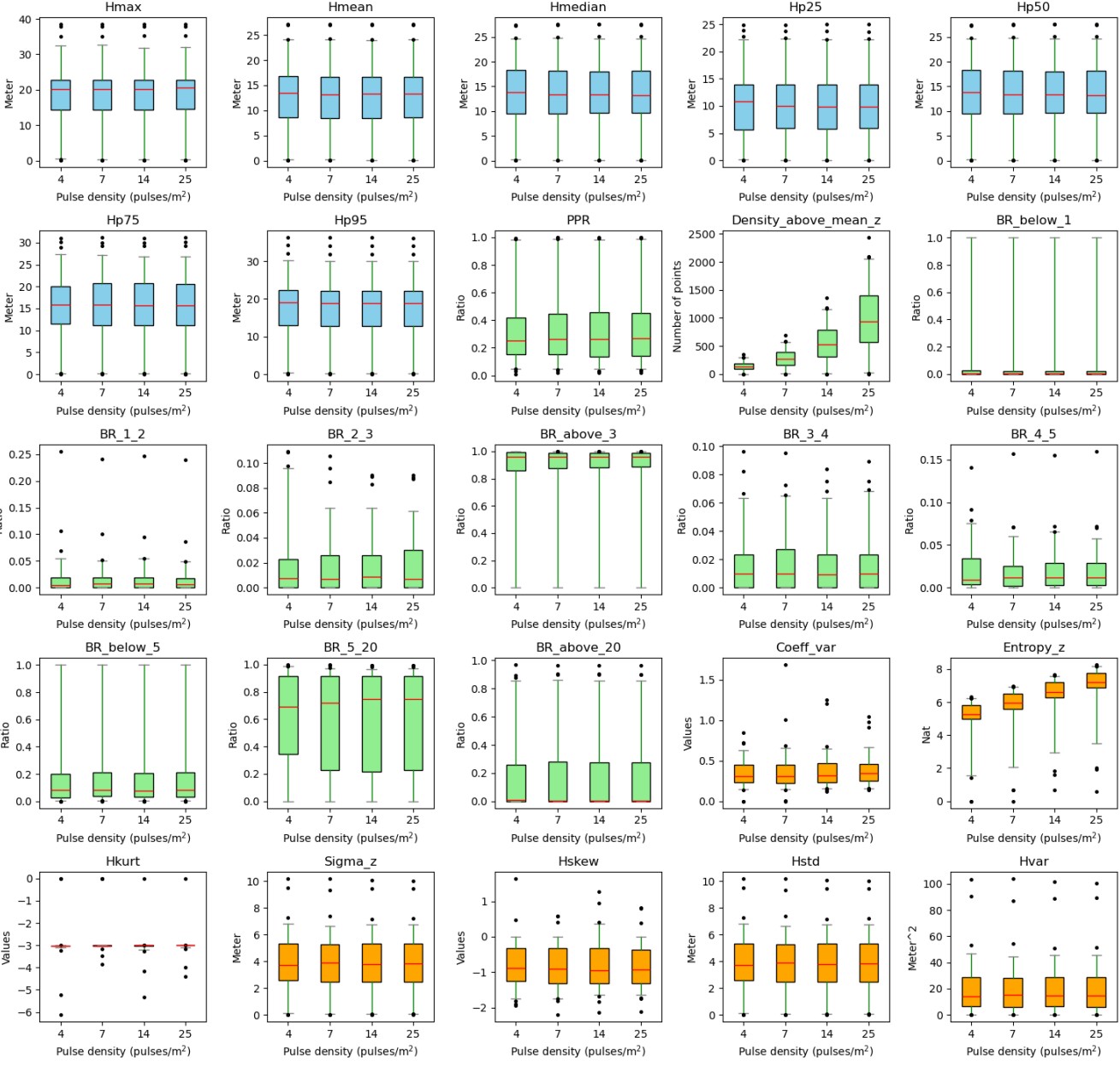

Figure B5. Robustness of vegetation metrics in woodland habitats. Twenty-five LiDAR metrics (blue:
vegetation height metrics, green: vegetation cover metrics, orange: vegetation structural variability
metrics) were calculated with different pulse densities across 100 plots of 10 × 10 m resolution in
woodland habitats in the Netherlands. Pulse densities were systematically down-sampled based on their
GPS time from the original AHN4 dataset to the pulse density of AHN3 and two lower pulse densities
(i.e. 1/2 and 1/4 of the pulse density of AHN3 to represent AHN2 and AHN1, respectively). Boxes
represent the interquartile range, horizontal red lines the medians, whiskers extend to the 5th and 95th
percentiles, and outliers are plotted as dots. See Table 3 for metric explanations.

## Appendix C

## Background

Since the methods/algorithms used in the pre-classification of the AHN datasets are unclear (no specific documents or information are publicly available) and differences in pre-classification methods between AHN datasets can potentially lead to some biases in vegetation change detection (Fareed et al., 2023; Wu et al., 2019), we performed a preliminary assessment of the effect of terrain filtering on vegetation change detection across AHN datasets (i.e. AHN2–AHN4).

## Study area

The study area for this analysis is in the Amsterdam Water Supply Dunes (AWD), which is a 34 km$^2$ dune ecosystem in the west of Amsterdam, stretching 8 km along the Dutch North Sea coast with a width varying from 1.5 to 5 km. The AWD area is dominated by various dune habitats, including shifting white dunes, fixed coastal dunes with herbaceous vegetation, dunes with sea-buckthorn formations, wooded dunes and humid dune slacks (Kissling et al., 2024b). Vegetation types include grasses (46 %), scrublands (22 %), forests (21 %), sand (6 %) and other low vegetation. To evaluate the impact of varying ground point classification approaches (for AHN2, AHN3, and AHN4) on derived LiDAR vegetation metrics, we selected three sample plots within the AWD area to conduct our analysis. We selected three sample areas (1 km × 1.3 km each) for this analysis, and the specific locations of each sample plots are: Area 1 (5.437882°E, 52.304127°N), Area 2 (5.480002°E, 52.278998°N), and Area 3 (5.501239°E, 52.289103°N).

## Methods

First, we computed 25 LiDAR-derived vegetation metrics using the pre-classified AHN datasets (class "unclassified" as in the main text. Second, we applied a filtering algorithm with identical parameter settings to the original multi-temporal AHN point clouds to reclassify the terrain and vegetation points consistently across AHN2–AHN4. We then derived the same 25 LiDAR metrics using the reclassified data, following the same workflow applied to the pre-classifications. All LiDAR metrics were derived and compared at a 10 m resolution. To further assess the differences in LiDAR-derived vegetation metric change across multi-temporal datasets, we conducted pairwise comparisons between AHN2 and AHN3, and between AHN3 and AHN4. The differences (delta metrics) were calculated by subtracting the vegetation metrics of the earlier datasets from those of the later ones (i.e. subtracting AHN2 from AHN3 and subtracting AHN3 from AHN4). The height of non-ground points was normalized using the height of the lowest point within each 1 m × 1 m grid cell (in line with the Laserfarm workflow). The resulting vegetation metrics were first exported as GeoTIFF files with a 10 m resolution, after which pixel-wise subtraction was performed.

We used an iterative grid-based filtering approach to segment terrain (i.e. ground) points from
raw LiDAR point clouds, enabling efficient separation of vegetation and ground points in the dune
environments. This filtering approach consists of four steps:

**Step 1: Preprocessing**

This step mainly removes the outliers of the original point cloud of AHN datasets. The statistical
outlier removal (SOR) was employed to remove noise points with the method proposed in Rusu et al.
(2008). Suppose $P$ is a set of 3D points, and for each query point $p_{query} \in P$, $\bar{d}$ is the mean distance of
a query point to its $k$ nearest neighbors. For all points in $P$, the mean distance and standard deviation of
the distances of their $k$ nearest neighbors are then determined. Only those points are kept which have
distances that are close to the mean distance of the closest neighbours, using Equation (1).

$$P^k = \{p_q \in P \mid (\mu_k - \alpha\sigma_k) \leq \bar{d} \leq (\mu_k + \alpha\sigma_k)\} \tag{1}$$

Here, $\alpha$ is a density threshold coefficient, and $\mu_k$ and $\sigma_k$ are the mean and standard deviation of
the distance from a query point to its $k$ closest neighbors. $P^k$ is the point set that is kept, i.e. after
removing the outliers.

**Step 2: Grid initialization**

The original 3D point cloud of the AHN is divided into a virtual grid layer, starting with a coarse
resolution. The indices of the grids are calculated using Equation 2.

$$n^i = \frac{P^i - P^i_{min}}{Size_g^i} (i \in x, y, z) \tag{2}$$

Here, $P^i$ is the coordinates of a point and $Size_g^i$ is the grid size.

**Step 3: Elevation interpolation**

For each grid cell in the bottom layer, elevation $E_g$ is interpolated using a distance-weighted
average of points within the grid using Equation 3.

$$E_g = \frac{\sum E_p (\frac{L}{\sqrt{2}} - D_g)}{\sum (\frac{L}{\sqrt{2}} - D_g)} \tag{3}$$

Here, $L$ is the grid size, $E_p$ is the elevation of a point, and $D_g$ is the distance from the point to the
geometric centre of the grid.

**Step 4: Iterative refinement**

The generated grids are iteratively subdivided by halving the grid size per iteration until reaching
the minimum grid size. For the points that exceeding a height threshold above the interpolated terrain
elevation are classified as vegetation points.

Finally, the original points are classified into terrain (i.e. ground) points and vegetation point
categories. The classified vegetation and terrain points are applied to the computation of the LiDAR
vegetation metrics. The parameter settings in this workflow were: minimum grid size: 1 m; maximum
grid size: 15 m; height threshold: 0.5 m.

**Results and Conclusions**

Our results revealed that the differences between the vegetation changes generated from point clouds
using the AHN pre-classification and using a consistent terrain filtering method across the AHN2–AHN4
datasets is negligible. The only exceptions were the pulse penetration ratio ("PPR"), the coefficient of
variation of vegetation height ("Coeff_var"), and the Shannon index ("Entropy_z"), where small
differences were observed (Fig. C1–C3). This analysis thus provides first insights into the reliability of
the pre-classification of the AHN datasets when calculating vegetation change. Conditional on those
results, we conclude that most LiDAR metrics based on the pre-classifications of AHN (AHN2–AHN4)
datasets are reliable, with only a few vertical variability metrics showing a detectable effect of potential
differences in the ground classification methods between AHN2–AHN4 datasets. It should be noted that
we conducted this assessment only in the Dutch coastal dunes, and similar assessments can be done across
different sites and different habitats in future studies for a more comprehensive understanding on this
topic.

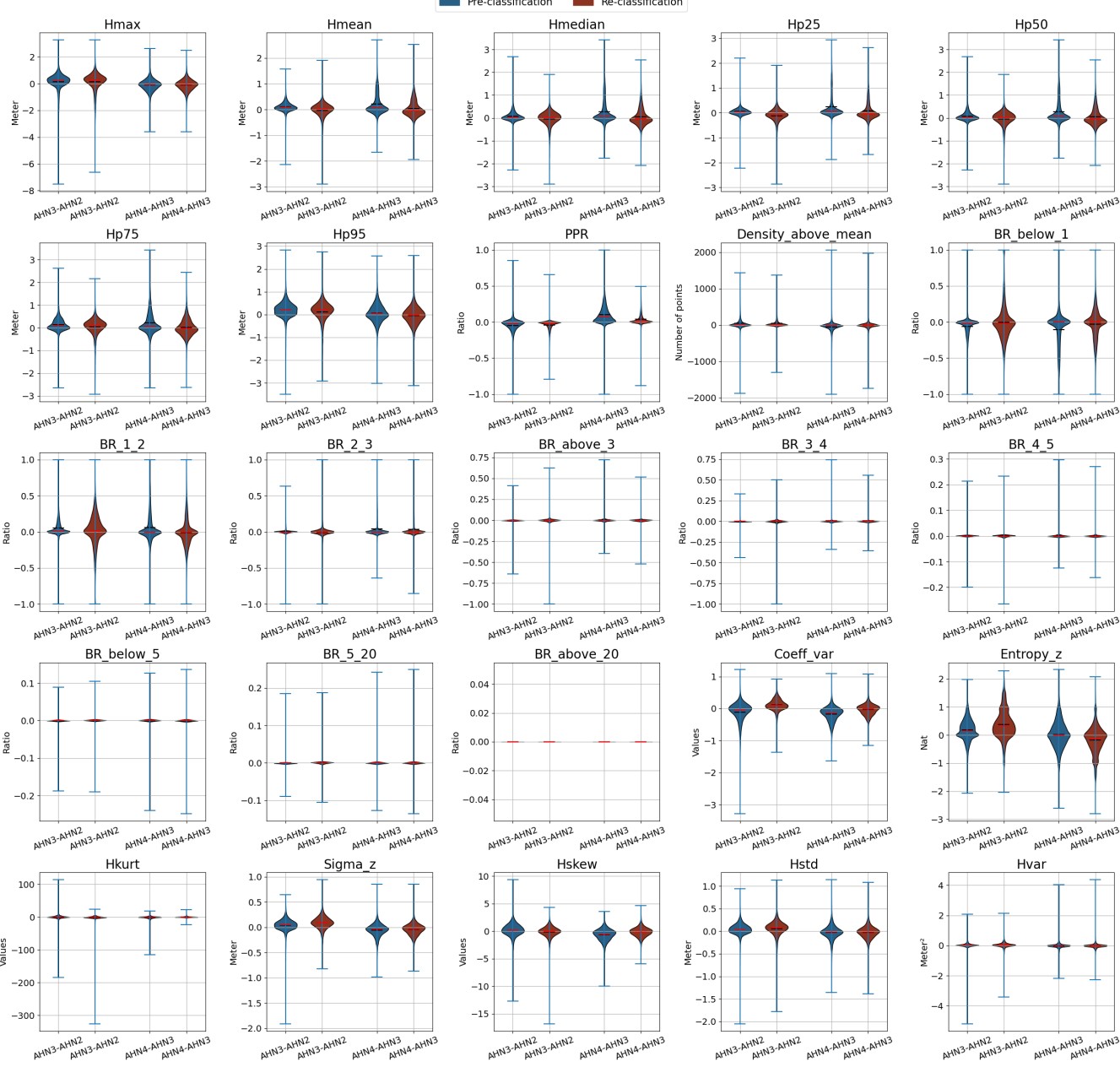

Figure. C1 Pixel-wise comparisons of LiDAR-derived vegetation changes from Area 1 using the pre-classifications from the AHN2–AHN4 datasets (blue) vs. those using a consistent terrain filtering method across the three AHN datasets (red). The total number of pixels in Area 1 is 13,416 (n = 13,416).

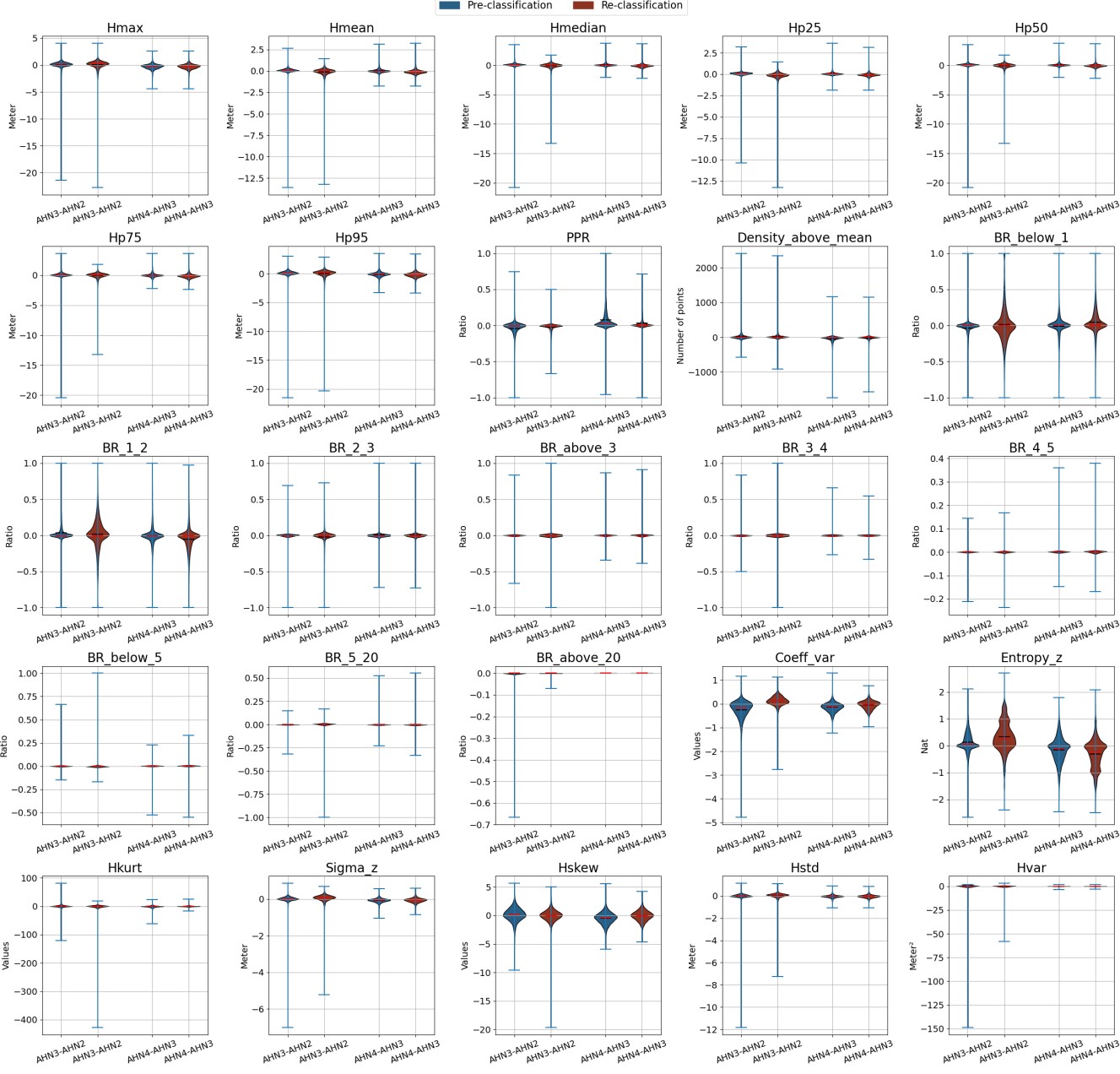

Figure. C2 Pixel-wise comparisons of LiDAR-derived vegetation changes from Area 2 using the pre-classifications from the AHN2–AHN4 datasets (blue) vs. those using a consistent terrain filtering method across the three AHN datasets (red). The total number of pixels in Area 2 is 13,416 (n = 13,416).

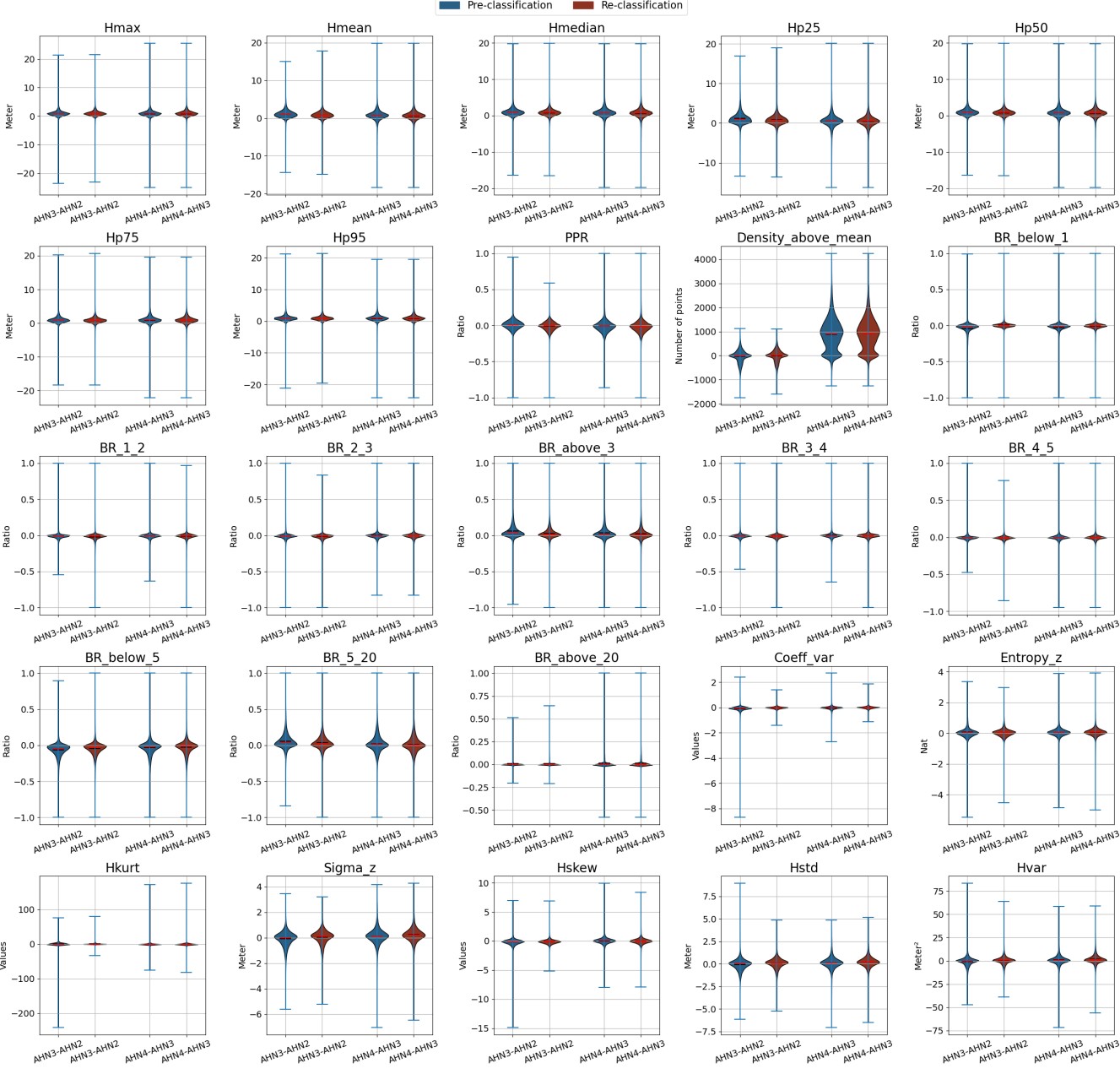

Figure. C3 Pixel-wise comparisons of LiDAR-derived vegetation changes from Area 3 using the pre-classifications from the AHN2–AHN4 datasets (blue) vs. those using a consistent terrain filtering method across the three AHN datasets (red). The total number of pixels in Area 3 is 13,416 (n = 13,416).

## Author contributions

**Yifang Shi**: Conceptualization, Data curation, Formal analysis, Methodology, Validation, Visualization, Writing – original draft, Writing – review & editing. **Jinhu Wang**: Formal analysis, Validation, Visualization, Writing – review & editing. **W. Daniel Kissling**: Conceptualization, Investigation, Funding acquisition, Project administration, Supervision, Writing – review & editing.

## Competing interests

The contact author has declared that none of the authors has any competing interests.

## Acknowledgements

We thank Fabian Fischer and one anonymous referee for stimulating and constructive comments on an earlier draft manuscript. We acknowledge funding support from the European Commission (MAMBO project grant number 101060639) and the Netherlands eScience Center (grant number ASDI.2016.014). We thank Francesco Nattino and Meiert W. Grootes from the Netherlands eScience Center for leading the development of the Laserfarm workflow through the project "eScience infrastructure for Ecological applications of LiDAR point clouds" (eEcoLiDAR) (Kissling et al., 2017). We further thank Francesco Nattino for making a new release of the Laserchicken software (https://github.com/eEcoLiDAR/laserchicken/issues/190). The development of the data products was also supported by LifeWatch ERIC (https://www.lifewatch.eu/), an European research infrastructure consortium with focus on biodiversity and ecosystem research. We acknowledge the computing resources provided by SURF, the Dutch national facility for information and communication technology (https://www.surf.nl/).

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
