# Peer review of "Multi-temporal high-resolution data products of ecosystem"

_Earth System Science Data, 2024_

## Author Comment (AC1)

**Reviewer 1**

The article presents a large dataset of 25 ecosystem structure metrics for the entire area of the Netherlands at 10 m resolution. The metrics are calculated from openly available airborne laser scanning (ALS) data and across multiple ALS campaigns, making it a highly valuable dataset to assess ecosystem dynamics. The article is well-written, with great attention to detail – I particularly commend the tables and figures that present a lot of information without feeling overly complex. The paper also follows a nice logical flow, from presenting the ALS pipeline, to detailed justifications for the derived metrics, to 2 sample case studies that have high relevance for applied research (changes in ecosystem structure + a comparison of structural indices across different ecosystem types). The resulting product provides insight into high resolution ecosystem change over at least 15 years from 2007-2022 (Note that I am not counting the first ALS campaign which likely does not reach minimum quality standards for ecological analysis). It thus has great potential to become a standard tool both for practitioners and researchers interested in ecosystems in the Netherlands. It could also become a nice reference dataset for similar efforts at larger scale.

**Response**: Thank you very much for your positive evaluation. We appreciate your valuable comments which have helped us to improve the quality of the manuscript. We have thoroughly addressed your comments one by one. See details below.

I will now provide my main comments, with line-by-line comments following below.

**Main comment 1: Robustness of pipeline to pulse density / leaf phenology**

The main issue where the authors have not (yet) convinced me is whether comparisons in time between different AHN surveys are robust to acquisition properties. Large disturbances (e.g. clear cutting, logging) will obviously be visible and can be separated from noise, but how about growth or smaller disturbances? The authors mention at the end of the introduction that intercomparisons between different instruments and scanning conditions could lead to considerable errors, but then do not really provide means to assess the sensitivity of the products or to correct for some of these problems. This is important, because the surveys of AHN1 and even AHN2 differ strongly from AHN3-5 in terms of pulse density, and even the more recent high quality surveys may differ in leaf phenology. E.g., a scan in April will likely already measure some vegetation in early leaf-on conditions, and this might create bias compared to a scan in December.

Overall, I see three points that I would like the authors to address:

**1. Sensitivity analysis:** The authors should provide a sensitivity analysis of the pipeline with regard to pulse density, e.g., how much do the inferred 25 metrics change when the pulse density of a high quality scan (AHN4-5) is degraded to levels of AHN1 or AHN2. This can then be used to provide clear bounds on what is ecologically interpretable. The simplest way to do

this would be to a) use the site in case study 1, degrade the point cloud from AHN5 and check robustness of the metrics, and b) to repeat case study 2, with the exact same sites and metrics, but with AHN1 or AHN2. Is the analysis reproducible for ecosystems with low vegetation and small differences between them? I do not expect all metrics to be perfectly stable for this study to be published, but having an estimate of uncertainty for all of them would be key. Note that I would degrade pulse density (the number of shots) and not point density (the number of shots + returns) to accurately simulate a lower-quality scan.

Response: Thank you for your insightful comment. We agree that the robustness of LiDAR metrics against varying pulse density is crucial for the interpretation and usage of the data products. We have therefore conducted a comprehensive sensitivity analysis on the robustness of the generated 25 LiDAR metrics based on different pulse densities. Note that for the four completed AHN surveys only the AHN3 and AHN4 provide pulse information (e.g. "return number", "number of returns") in the point cloud, whereas the AHN1 and AHN2 do not provide such information. Considering that different pulse densities can potentially have different effects on LiDAR metrics in different habitat types, we have performed this sensitivity analysis in five major habitat types (i.e. dunes, marshes, grasslands, shrublands, and woodlands) within Natura 2000 sites in the Netherlands. For each habitat type, we randomly selected 100 sample plots (10  $m \times 10$  m, 500 plots in total) in which Hp95 is not NA (i.e. assuming that vegetation occurs in the plots). The detailed methodology for plot selection is provided in Appendix A. For each sample plot, the pulse density (i.e. the density of first return points) of AHN4 was systematically down-sampled to the same pulse density as AHN3, and then to 1/2 of the pulse density of AHN3 (assuming comparability with AHN2), and to 1/4 of the pulse density of AHN3 (assuming comparability with AHN1). For systematic down-sampling, we used a methodology that we recently developed (see Appendix B of Kissling et al. 2024), i.e. the first return points were first sorted according to their GPS acquisition time (from earliest to latest) and then down-sampled to the different densities. For instance, for woodlands, we have down-sampled pulse density from 25 pulses/m2 (AHN4) to 14 pulses/m2, 7 pulses/m2, and 4 pulses/m2, respectively. We then calculated the 25 LiDAR metrics for the original AHN4 point cloud and for the down-sampled point clouds. Our analysis revealed that almost all LiDAR-derived vegetation metrics in all habitats are robust to varying pulse densities at 10 m resolution, even when calculated with strongly down-sampled pulse densities of  $\leq 4$  pulses/m2 (see Figure B1–B5 in Appendix B). The exception was canopy cover ("Density above mean z") and Shannon index ("Entropy z") which markedly decreased with lower pulse densities in all habitat types, and the coefficient of variation of vegetation height ("Coeff var") in grasslands and shrublands (Appendix B). Some metrics in grasslands also showed larger variability with down-sampled pulse densities.

This sensitivity analysis provides comprehensive insights into the effects of pulse densities on metric calculations across different habitat types and helps to make our data ecologically interpretable. From this we derived usage notes for users to interpret the LiDAR metrics. We have included the sensitivity analysis and our interpretation into the revised manuscript under Sect. 3.4 "Limitations and usage notes". See Sect. 3.4.4 "Sensitivity analysis" and Appendix B

for more details. We have also added a few sentences in the discussion to address this point. See lines 703–712.

**2. Leaf phenology:** If possible, the authors should provide a timestamp/acquisition time for every 10 m x 10 m pixel. There are vector files with information on flight lines available for AHN2-5 online, so maybe these could be rasterized? Some of the layers are likely incomplete, but having information on flight time for the pixels down to the month would make the dataset very valuable.

**Response**: We agree that providing a timestamp raster layer at 10 m resolution would be beneficial for comparing the time of acquisition for each grid cell among the datasets and generated properties. We have now processed the flight line information to timestamp raster layers for both AHN3 and AHN4 at 10 m resolution across the whole Netherlands. For both AHN3 and AHN4 surveys, a flightline vector layer is available with complete flight year/month/date information across the country. Note that the flightline layer of AHN2 is not complete and only a small portion of available flightlines has information on flight year and month. Hence, we eventually did not include the AHN2 flightlines in the processing. For the ongoing AHN5 survey, the flight timestamp raster layer can be included in the future. For AHN3 and AHN4, we first downloaded the flightline vector layers from https://www.ahn.nl/dataroom, and then generated a buffer zone around the flightlines using the function "Buffer" in ArcGIS Pro with the setting of a distance (on both sides of each flightline) of 300 m for AHN3 and 700 m for AHN4, and dissolved the neighboring buffer zones if they had the same flight time. The distance of the buffer zone was set based on the distance between two flightlines for the target AHN survey. We then rasterized the generated buffer zone polygons into raster layers at 10 m resolution. For areas with multiple flightlines overlapping, we assigned the latest flight date to the raster pixel to be consistent with the flight year maps provided by AHN (see Fig. 1). We suggest that users take the surrounding pixel information into account when investigating overlapping areas. We make the generated timestamp layers for AHN3 and AHN4 available in the same data repository as the data products.

We have added related information in section 3.3 Auxiliary data, see lines 354–369.

**3. Quantitative guidelines:** Finally, the paper should provide clear **quantitative guidelines** for researchers or practitioners on what kind of differences are interpretable ecologically. E.g., if I notice a change in height of 1m between 2017 and 2022, is this a real height change, or does this fall within the uncertainty due to laser instrumentation/DTM derivation? Vegetation growth can be slow (0.1-0.5 m/ year), so it is important to be able to separate noise/artefacts from actual change. Cf. also the second main comment.

**Response**: We agree that small height changes (e.g. within 0.5-1 m) can be difficult to distinguish from noises or uncertainties introduced by systematic errors or DTM derivation. Given the vertical accuracy of AHN2–AHN4 (i.e. 5–15 cm), classification related errors, and the potential influence of acquisition time of the datasets, we suggest that small vegetation changes (e.g. less than 0.5-1 m) should be interpreted with caution. Such small height changes can be

influenced by vertical height uncertainties, low vegetation points being wrongly classified as ground points, or differences in leaf phenology due to varying data acquisition times rather than representing real vegetation changes. When comparing vegetation changes between the AHN3 and AHN4 metrics, users can make use of the flight time raster layers to take vegetation phenology differences into account. Based on our sensitivity analysis, we also suggest that users should be aware that some LiDAR metrics from open and heterogeneous habitats such as grasslands and shrublands might be less robust to varying point and pulse densities than those from dunes, marshes and woodlands.

We have added this information in the revised manuscript under Sect. 3.4 "Limitations and usage notes". See lines 525–534.

**Main comment 2: Independent comparison and NA values**

I carried out a quick comparison with our own laser scanning pipeline, which we have previously tested for robustness (Fischer et al., 2024, Methods in Ecology and Evolution (https://doi.org/10.1111/2041-210X.14416). I will call this the LAStools pipeline, and the author's pipeline the Laserfarm pipeline. I have uploaded the products of this comparison to Zenodo so that the authors can compare it to their results: https://zenodo.org/records/14722001.

I only carried out a simple comparison: I compared the 95th percentile of canopy height at 10 m resolution from two CHMs produced via the LAStools pipeline ("chm\_lspikefree.tif" and "chm\_tin.tif") with the Laserfarm hp95 product at the site in case study 1. The main findings are:

**1. NA values**: The products in the Laserfarm hp95 sometimes seem to have a considerable amount of NA values. Unfortunately, these NA values are not consistent across AHN surveys, so comparisons of canopy height change may vary depending on how these NA values are dealt with across surveys: when I ignored these differences and simply calculated the difference in canopy height means at the study site, I would get a height loss of -3.84 m from AHN2 to AHN3, a height increase of 0.45 m from AHN3 to AHN4, and then again a loss of -0.57 m from AHN4 to AHN5. When only considering areas that were not NA in any of the surveys, the height changes were as follows: -3.30 m, -0.03 m, -0.12 m, so differences of up to 0.5 m. The authors should either try to remove the NA values consistently from all products (this should be possible, as shown by the products I derived with the LAStools pipeline), or provide a mask and a clear guideline on how to deal with them. Cf. the attached pdf for a visualization of the NA values.